# DIGRAC: DIGRAPH CLUSTERING BASED ON FLOW IMBALANCE

## ABSTRACT

Node clustering is a powerful tool in the analysis of networks. We introduce a graph neural network framework to obtain node embeddings for directed networks in a self-supervised manner, including a novel probabilistic imbalance loss, which can be used for network clustering. Here, we propose *directed flow imbalance* measures, which are tightly related to directionality, to reveal clusters in the network even when there is no density difference between clusters. In contrast to standard approaches in the literature, in this paper, directionality is not treated as a nuisance, but rather contains the main signal. DIGRAC optimizes directed flow imbalance for clustering without requiring label supervision, unlike existing GNN methods, and can naturally incorporate node features, unlike existing spectral methods. Experimental results on synthetic data, in the form of directed stochastic block models, and real-world data at different scales, demonstrate that our method, based on flow imbalance, attains state-of-the-art results on directed graph clustering, for a wide range of noise and sparsity levels and graph structures and topologies.

## 1 INTRODUCTION

Revealing an underlying community structure of *directed* networks (*digraphs*) is an important problem in many application domains (Malliaros & Vazirgiannis, 2013), such as detecting influential social groups (Bovet & Grindrod, 2020) and analyzing migration patterns (Perry, 2003); node clustering in digraphs can be used for this community detection task (Palmer & Zheng, 2021). While most existing methods that could be applied to directed clustering use local edge densities as main signal and directionality as additional signal, we argue that even in the absence of any edge density differences, directionality can play a vital role in directed clustering as it can reveal latent properties of network flows. Therefore, instead of finding relatively dense groups of nodes in digraphs which have a relatively small amount of flow between the groups, as in (Girvan & Newman, 2002; Newman, 2006; Leskovec et al., 2008; Leicht & Newman, 2008; Chen & Baker, 2021; Jia et al., 2017), our main goal is to recover clusters with *strong and imbalanced flow* among them, in the spirit of (Cucuringu et al., 2020a; Laenen & Sun, 2020), where directionality (i.e, edge orientation) is the main signal.

In contrast to standard approaches focusing on edge density, here edge directionality is not a nuisance but the main piece of information to uncover the latent structure. The underlying intuition is that homogeneous clusters of nodes form *meta-nodes* in a *meta-graph*, with the meta-graph directing the flow between clusters; directed core-periphery structure is such an example (Elliott et al., 2020). Figure 1(a) is an example of flow imbalance between two clusters, here on an unweighted network for simplicity: while 75% of the edges flow from the *Transient* cluster to the *Sink* cluster, only 25% flow in the other direction. As a real-world example, Figure 1(b) shows the strongest flow imbalances between clusters in a network of US migration flow (Perry, 2003); most edges flow from the red cluster (label 1) to the blue one (label 2). Figures 1(c-d) show examples on a synthetic meta-graph.

While most competitive state-of-the-art methods proposed for node clustering in digraphs mainly rely on edge density and overlook the role of directionality, those that do lay emphasis on directionality, usually by incorporating cut imbalance among clusters, are usually spectral methods, for which incorporating features is non-trivial, or GNN methods that require labeling information. In general, community detection methods for networks are not able to find communities in networks in which the sole structure lies in the directionality of edges. An exception is InfoMap (Rosvall & Bergstrom, 2008), a method for community detection in networks which uses directed random walks; however, it still relies on some edge density information within clusters.

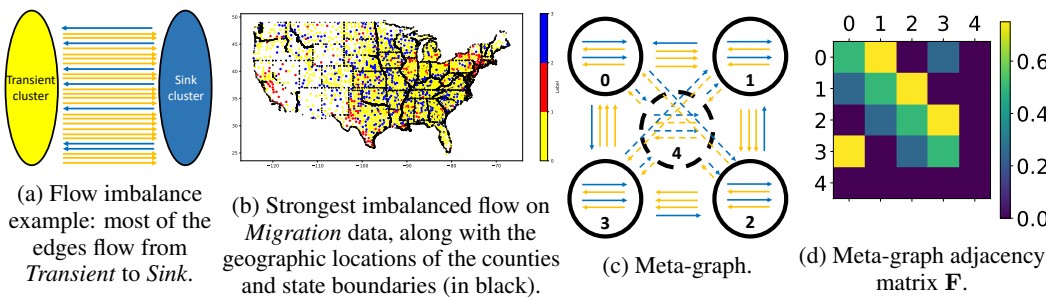

Figure 1: Visualization of cut flow imbalance and meta-graph: (a) 75% of edges flow from Transient to Sink, while 25% of edges flow in the opposite direction; (b) top pair imbalanced flow on *Migration* data (Perry, 2003): most edges flow from red (1) to blue (2); (c) & (d) are for a Directed Stochastic Block Model with a cycle meta-graph with ambient nodes, for a total of 5 clusters. 75% of the edges flow in direction $0 \rightarrow 1 \rightarrow 2 \rightarrow 3 \rightarrow 0$, while 25% flow in the opposite direction. Cluster 4 is the ambient cluster. In (c), dashed lines indicate flows with random equally likely directions; these flows do not exist in the meta-graph adjacency matrix $\mathbf{F}$. For (d), the lighter the color, the stronger the flow.

In this paper, we introduce a graph neural network (GNN) framework, denoted DIGRAC, to obtain node embeddings for clustering digraphs (potentially weighted, possibly with self-loops, but no multiple edges). In a self-supervised manner, a novel *probabilistic imbalance loss* is proposed to act on the digraph induced by all training nodes. The global imbalance score, one minus whom is the self-supervised loss function, is aggregated from pairwise normalized cut imbalances. The method is end-to-end in combining embedding generation and clustering without an intermediate step. To the best of our knowledge, this is the first GNN method that derives node embeddings for digraphs that directly maximizes flow imbalance between pairs of clusters. For the implementation of the framework, we devise a directed mixed path aggregation scheme that we abbreviate DIMPA, which is simple and effective. Other aggregation schemes which detect direction information could be used. With an emphasis on the use of a direction-based flow imbalance objective, experimental results on synthetic data and real-world data at different scales demonstrate that our method can achieve state-of-the-art performance for a wide range of network densities and topologies. Compared to its competitors, for synthetic data, our method achieves superior performance (with respect to the Adjusted Rand Index (ARI) (Hubert & Arabie, 1985)); for real-world data, our method clearly outperforms them, using imbalance scores as outcome success measures.

DIGRAC's main novelty is the ability to cluster based on direction-based flow imbalance, instead of using classical criteria such as maximizing relative densities within clusters. Compared with prior methods that focus on directionality, DIGRAC can easily consider node features and also does not require known labels. DIGRAC complements existing approaches in various aspects: (1) Our results show that DIGRAC complements classical community detection by detecting alternative patterns in the data, such as meta-graph structures, which are otherwise not detectable by existing methods. This aspect of detecting novel structures in directed graphs has also been emphasized in (Cucuringu et al., 2020a). (2) DIGRAC complements existing spectral methods, through the possibility of including exogenous information, in the form of node-level features or labels, thus borrowing their strength. (3) DIGRAC complements existing GNN methods by introducing an imbalance-based objective.

DIGRAC's applicability extends beyond settings where the input data is a digraph; for example, with time series data as input, the digraph construction mechanism can accommodate any procedure that encodes a pairwise directional association between the corresponding time series, such as lead-lag relationships and Granger causality (Shojaie & Fox, 2021), with applications such as in the analysis of information flow in brain networks (Dhamala et al., 2008), biology (Runge et al., 2019), finance (Bennett et al., 2021; Sornette & Zhou, 2005) and earth sciences (Harzallah & Sadourny, 1997).

Our method for extracting flow-driven clusters in digraphs based on higher-order meta-graphs can facilitate tasks in time series analysis, ranking, and anomaly detection, as it allows one to extrapolate from *local* pairwise (directed) interactions to a *global* structure inference, in the high-dimensional low signal-to-noise ratio regime. As another use case, consider a social network where a set of fake accounts $\mathcal{A}$ have been created, which are likely to target another subset $\mathcal{B}$ of real accounts by sending them messages. Most likely, there would be many more messages from $\mathcal{A}$ to $\mathcal{B}$ than from $\mathcal{B}$ to $\mathcal{A}$,

hinting that $\mathcal{A}$ is most likely comprised of fake accounts. In ranking applications, where the match results (or preference relationships between items) are encoded in a digraph, DIGRAC would allow for the detection of subsets of players where the relative strength of players in $\mathcal{A}$ is superior to that of players in $\mathcal{B}$, thus facilitating the extraction of partial rankings (Cucuringu, 2016).

**Main contributions.** Our main contributions are as follows. • (1) We propose a GNN framework for self-supervised end-to-end node clustering on digraphs, which are possibly attributed and weighted, explicitly taking into account the directed flow imbalance. • (2) We propose a probabilistic version of the global imbalance score to serve as a self-supervised loss function. To the best of our knowledge, this is the first method directly maximizing cut flow imbalance for node clustering in digraphs using GNNs. This new imbalance score is the key novelty in this paper. • (3) We extend our method to the semi-supervised setting when label information is available.

**Paper outline.** Section 2 reviews existing work. Section 3 introduces our DIGRAC method. Section 4 validates its strength through extensive experiments at different scales, on synthetic and real-world data. Section 5 draws conclusions and discusses limitations and future works. In Supplementary Information (SI), Section A expands on variants of our model; Section B discusses implementation details; Sections C, D & E provide additional results.

Anonymized codes and preprocessed data are available at `https://anonymous.4open.science/r/1b728e97-cc2b-4e6a-98ea-37668813536c`.

## 2 RELATED WORK

Directed clustering has been explored by spectral and GNN-based methods. Satuluri & Parthasarathy (2011) performs directed clustering that hinges on symmetrizations of the adjacency matrix, but is not scalable as it requires large matrix multiplication. Rohe et al. (2016) proposes a spectral co-clustering algorithm for asymmetry discovery that relies on in-degree and out-degree. Whenever direction is the sole information, such as in a complete network with lead-lag structure derived from time series Bennett et al. (2021), a purely degree-based method cannot detect the clusters. While Zhang et al. (2021a) produces two partitions of the node set, one based on out-degree and one based on in-degree, here we produce a partition that simultaneously takes both directions into account. Cucuringu et al. (2020a) seeks to uncover clusters characterized by a strongly imbalanced flow circulating among them, based on eigenvectors of the Hermitian matrix $(\mathbf{A} - \mathbf{A}^T) \cdot i$, where $\mathbf{A}$ is the (normalized) adjacency matrix and $i$ the imaginary unit. Zhang et al. (2021b) builds upon Cucuringu et al. (2020a); Mohar (2020) and introduces a complex Hermitian matrix that encodes undirected geometric structure in the magnitude of its entries, and directional information in their phase. Laenen & Sun (2020) uncovers higher-order structural information among clusters in digraphs, while maximizing the imbalance of the edge directions, but its definition of the flow ratio restricts the underlying meta-graph to a path.

Ma et al. (2019) and Palmer & Zheng (2021) introduce *directed graph Laplacians*, but these methods are only applicable to strongly connected digraphs, which is hardly the case in sparse networks arising in applications. Monti et al. (2018) utilizes convolution-like anisotropic filters based on local subgraph structures (motifs) for semi-supervised node classification tasks in digraphs, but relies on pre-defined structures and fails to handle complex networks. DGCN Tong et al. (2020b) uses first and second order proximity, constructs three Laplacians, but the method is space and speed inefficient. DiGCN Tong et al. (2020a) simplifies DGCN, builds a directed Laplacian based on PageRank, and aggregates information dependent on higher-order proximity. In addition, Zhang et al. (2021b); Monti et al. (2018); Tong et al. (2020b;a) all require known labels, which are not generally available for real-world data. Satuluri & Parthasarathy (2011); Rohe et al. (2016); Cucuringu et al. (2020a); Palmer & Zheng (2021); Laenen & Sun (2020) could not trivially incorporate node attributes or node labels. In contrast, we propose an efficient GNN-based method that maximizes a probablistic flow imbalance objective, in a self-supervised manner, and which can naturally analyze attributed weighted digraphs.

InfoMap by Rosvall & Bergstrom (2008) assumes that there is a "map" underlying the network, similar to a meta-graph in DIGRAC. However, InfoMap aims to minimize the expected description length of a random walk and is recommended for networks where edges encode patterns of movement among nodes. While related to DIGRAC, InfoMap still relies on some signals within modules.

To avoid potential misunderstanding, we briefly mention some related papers that we are aware of but do not compare against in our experiments in the main text. Lancichinetti et al. (2011) proposes the OSLOM method, which is very flexible but is based on a density heuristic and hence a comparison to DIGRAC on networks without density signal would not be fair to begin with. Dugué & Perez

(2015) introduces directionality in the Louvain algorithm. This algorithm optimizes a modularity-type function that compares the number of edges within communities to the expected number of edges under a specified model. It is thus an approach that aims to find denser than expected groups of vertices. When all groups have the same density, as in our synthetic data sets, and the only structure lies in the directionality of the edges, this method simply cannot be expected to perform well. Traag et al. (2019), the Leiden algorithm, also builds on the Louvain algorithm, again optimizing a modularity-type function that compares the number of edges within communities to the expected number of edges under a specified model. It is a powerful method for that task, but cannot be fairly compared to DIGRAC which is tailored to find imbalances.

We also do not compare DIGRAC against graph pooling methods (Bianchi et al., 2020), which are inspired by pooling in CNN and developed to discard information which is superfluous for the task at hand. A partition of the nodes which can be interpreted as clustering is only a byproduct of graph pooling. Although not often stated explicitly, graph pooling methods are usually developed only for undirected networks. While graph matching as in Xu et al. (2019a;b); Chowdhury & Needham (2021) and Xu (2020) can be viewed as a clustering method of networks, matching the graph of interest to a disconnected graph by connecting each node in the observed graph with an isolated node of the disconnected graph, this approach is not developed for directed networks. The underlying idea of these papers is complementary to the meta-graph idea which underpins DIGRAC; in the meta-graph, the components are connected, and estimating the directionality of these connections is the main focus. Hence this work addresses a very different task. We emphasize that these are all excellent methods, but they address different objectives and tasks. As confirmed by our experiments in SI E, comparing these methods to DIGRAC is not appropriate. DIGRAC is tailored to detect an imbalance signal in directed networks, and such a signal cannot be present in an undirected network. As it is based on imbalance, DIGRAC will not be able to detect a signal in an undirected network, thus rendering it not applicable to undirected networks.

## 3 THE DIGRAC METHOD

**Problem definition.** Denote a (possibly weighted) digraph with node attributes as $\mathcal{G} = (\mathcal{V}, \mathcal{E}, w, \mathbf{X}_\mathcal{V})$, with $\mathcal{V}$ the set of nodes, $\mathcal{E}$ the set of directed edges or links, and $w \in [0, \infty)^{|\mathcal{E}|}$ the set of edge weights. $\mathcal{G}$ may have self-loops, but no multiple edges. The number of nodes is $n = |\mathcal{V}|$, and $\mathbf{X}_\mathcal{V} \in \mathbb{R}^{n \times d_{\text{in}}}$ is a matrix whose rows encode the nodes' attributes [1]. Such a network can be represented by the attribute matrix $\mathbf{X}_\mathcal{V}$ and the adjacency matrix $\mathbf{A} = (A_{ij})_{i,j \in \mathcal{V}}$, with $\mathbf{A}_{ij} = 0$ if no edge exists from $v_i$ to $v_j$; if there is an edge $e$ from $v_i$ to $v_j$, we set $A_{ij} = w_e$, the edge weight.

Digraphs often lend themselves to interpreting weighted directed edges as flows, with a meta-graph on clusters of vertices describing the overall flow directions; see Figure 1. A clustering is a partition of the set of nodes into $K$ disjoint sets (clusters) $\mathcal{V} = \mathcal{C}_0 \cup \mathcal{C}_1 \cup \cdots \cup \mathcal{C}_{K-1}$. Intuitively, nodes within a cluster should be similar to each other with respect to flow directions, while nodes across clusters should be dissimilar. In a self-supervised setting, only the number of clusters $K$ is given. In a semi-supervised setting, for each of the $K$ clusters, a fraction set $\mathcal{V}^{\text{seed}} \subseteq \mathcal{V}^{\text{train}} \subset \mathcal{V}$ of the set $\mathcal{V}^{\text{train}}$ of all training nodes are selected as seed nodes, for which the cluster membership labels are known before training. The goal of semi-supervised clustering is to assign each node $v \in \mathcal{V}$ to a cluster containing some known seed nodes, without knowledge of the underlying flow meta-graph. The corresponding self-supervised clustering task does not use seed nodes.

### 3.1 SELF-SUPERVISED LOSS FOR CLUSTERING

Our self-supervised loss function is inspired by Cucuringu et al. (2020a), aiming to cluster the nodes by maximizing a normalized form of cut imbalance across clusters. We first define probabilistic versions of cuts, imbalance flows, and probabilistic volumes. For $K$ clusters, the *assignment probability matrix* $\mathbf{P} \in \mathbb{R}^{n \times K}$ has as row $i$ the probability vector $\mathbf{P}_{(\mathbf{i},:)} \in \mathbb{R}^K$ with entries denoting the probabilities of each node to belong to each cluster; its $k^{\text{th}}$ column is denoted by $\mathbf{P}_{(:,k)}$.

- The **probabilistic cut** from cluster $\mathcal{C}_k$ to $\mathcal{C}_l$ is defined as

$$W(\mathcal{C}_k, \mathcal{C}_l) = \sum_{i,j \in \{1,\ldots,n\}} \mathbf{A}_{i,j} \cdot \mathbf{P}_{i,k} \cdot \mathbf{P}_{j,l} = (\mathbf{P}_{(:,k)})^T \mathbf{A} \mathbf{P}_{(:,l)}.$$

---

[1] If no attributes are available, one could use any feature matrix generated from $\mathbf{A}$, such as stacking the eigenvectors of the Hermitian matrix introduced in (Cucuringu et al., 2020a), to construct the feature matrix.

• The **imbalance flow** between clusters $\mathcal{C}_k$ and $\mathcal{C}_l$ is defined as

$$|W(\mathcal{C}_k, \mathcal{C}_l) - W(\mathcal{C}_l, \mathcal{C}_k)|, \quad \forall k, l \in \{0, \ldots, K-1\}.$$

For interpretability and ease of comparison, we normalize the imbalance flows to obtain an imbalance score with values in $[0, 1]$ as follows (we defer additional details to SI A.1).

• The **probabilistic volume** for cluster $\mathcal{C}_k$ is defined as

$$VOL(\mathcal{C}_k) = VOL^{(\text{out})}(\mathcal{C}_k) + VOL^{(\text{in})}(\mathcal{C}_k) = \sum_{i,j} (\mathbf{A}_{i,j} + \mathbf{A}_{j,i}) \cdot \mathbf{P}_{j,k}.$$

Then $VOL(\mathcal{C}_k) \geq W(\mathcal{C}_k, \mathcal{C}_l)$ for all $l \in \{0, \ldots, K-1\}$ and

$$\min(VOL(\mathcal{C}_k), VOL(\mathcal{C}_l)) \geq \max(W(\mathcal{C}_k, \mathcal{C}_l), W(\mathcal{C}_l, \mathcal{C}_k)) \geq |W(\mathcal{C}_k, \mathcal{C}_l) - W(\mathcal{C}_l, \mathcal{C}_k)|. \quad (1)$$

The imbalance term, which is used in most of our experiments, denoted CI$^{\text{vol\_sum}}$, is defined as

$$\text{CI}^{\text{vol\_sum}}(k, l) = 2 \frac{|W(\mathcal{C}_k, \mathcal{C}_l) - W(\mathcal{C}_l, \mathcal{C}_k)|}{VOL(\mathcal{C}_k) + VOL(\mathcal{C}_l)} \in [0, 1]. \quad (2)$$

The aim is to find a partition which maximizes the imbalance flow under the constraint that the partition has at least two sets, to capture groups of nodes which could be viewed as representing clusters in the meta-graph. The normalization by the volumes penalizes partitions that put most nodes into a single cluster. The range $[0, 1]$ follows from Eq. (1). Other variants are discussed in SI A.2.

To obtain a **global probabilistic imbalance score**, based on CI$^{\text{vol\_sum}}$ from Eq. (2), we average over pairwise imbalance scores of different pairs of clusters. Since the scores discussed are symmetric and the cut difference before taking absolute value is skew-symmetric, we only need to consider the pairs $\mathcal{T} = \{(\mathcal{C}_k, \mathcal{C}_l) : 0 \leq k < l \leq K-1, k, l \in \mathbb{Z}\}$. We consider a "*sort*" variant to select these pairs; concretely, we choose the pairs of clusters with the largest $\beta$ pairwise cut flow imbalance values, where $\beta$ is half of the number of nonzero entries in the off-diagonal entries of the meta-graph adjacency matrix $\mathbf{F}$ (if the meta-graph is known), or can be approximated otherwise. With $\mathcal{T}(\beta) = \{(\mathcal{C}_k, \mathcal{C}_l) \in \mathcal{T} : \text{CI}^{\text{vol\_sum}}(k, l) \text{ is among the top } \beta \text{ values}\}$, where $1 \leq \beta \leq \binom{K}{2}$, we set

$$\mathcal{O}^{\text{sort}}_{\text{vol\_sum}} = \frac{1}{\beta} \sum_{(\mathcal{C}_k, \mathcal{C}_l) \in \mathcal{T}(\beta)} \text{CI}^{\text{vol\_sum}}(k, l), \quad \text{and} \quad \mathcal{L}^{\text{sort}}_{\text{vol\_sum}} = 1 - \mathcal{O}^{\text{sort}}_{\text{vol\_sum}}, \quad (3)$$

as the corresponding loss function. For example, for a "*cycle*" meta-graph with three clusters and no ambient nodes, we have $\beta = 3$. When we consider a "*path*" meta-graph with three clusters and ambient nodes, we have $\beta = 1$. Definitions of meta-graph structures are discussed in Section 4.1. In SI A.3, additional variants for selecting pairs of clusters in $\mathcal{T}$ are considered, including an "*std*" variant based on hypothesis testing, and a "*naive*" variant which includes all pairs in $\mathcal{T}$. The corresponding scores and loss functions for these variants are defined analogously.

## 3.2 DIRECTED MIXED PATH AGGREGATION (DIMPA)

To implement DIGRAC, we devise a simple yet effective directed mixed path aggregation scheme, to obtain the probability assignment matrix $\mathbf{P}$ and feed it to the loss function, as a digraph extension of the successful *KernelGCN* method introduced by Tian et al. (2019). Thus, in order to build node embeddings, we capture local network information by taking a weighted average of information from neighbors within $h$ hops. To this end, we row-normalize the adjacency matrix, $\mathbf{A}$, to obtain $\overline{\mathbf{A}}^s$. Similar to the regularization discussed in Kipf & Welling (2016), we add a weighted self-loop to each node and normalize by setting $\overline{\mathbf{A}}^s = (\tilde{\mathbf{D}}^s)^{-1} \tilde{\mathbf{A}}^s$, where $\tilde{\mathbf{A}}^s = \mathbf{A} + \tau \mathbf{I}$, and the diagonal matrix $\tilde{\mathbf{D}}^s(i, i) = \sum_j \tilde{\mathbf{A}}^s(i, j)$, for a small value $\tau$, such as 0.5.

The $h$-hop **source** matrix is given by $(\overline{\mathbf{A}}^s)^h$. We denote the set of *up-to-$h$-hop* source neighborhood matrices as $\mathcal{A}^{s,h} = \{\mathbf{I}, \overline{\mathbf{A}}^s, \ldots, (\overline{\mathbf{A}}^s)^h\}$. Similarly, for aggregating information when each node is construed as a **target** node of a link, we carry out the same procedure for $\mathbf{A}^T$. We denote the set of up-to-$h$-hop target neighborhood matrices as $\mathcal{A}^{t,h} = \{\mathbf{I}, \overline{\mathbf{A}}^t, \ldots, (\overline{\mathbf{A}}^t)^h\}$, where $\overline{\mathbf{A}}^t$ is the row-normalized target adjacency matrix calculated from $\mathbf{A}^T$.

Next, we define two feature mapping functions for source and target embeddings, respectively. Assume that for each node in $\mathcal{V}$, a vector of features is available, and summarize these features in the input feature matrix $\mathbf{X}_{\mathcal{V}}$. The source embedding (the superscript $s$ stands for source) is given by

$$\mathbf{Z}_{\mathcal{V}}^s = \left( \sum_{\mathbf{M} \in \mathcal{A}^{s,h}} \omega_{\mathbf{M}}^s \cdot \mathbf{M} \right) \cdot \mathbf{H}_{\mathcal{V}}^s \in \mathbb{R}^{n \times d}, \tag{4}$$

where for each $\mathbf{M}$, $\omega_{\mathbf{M}}^s$ is a learnable scalar, $d$ is the dimension of this embedding, and $\mathbf{H}_{\mathcal{V}}^s = \mathbf{MLP}^{(s,l)}(\mathbf{X}_{\mathcal{V}})$. Here, the hyperparameter $l$ controls the number of layers in the multilayer perceptron (MLP) with ReLU activation; we fix $l = 2$ throughout. Each layer of the MLP has the same number $d$ of hidden units. The target embedding $\mathbf{Z}_{\mathcal{V}}^t$ is defined similarly, with $s$ replaced by $t$ in Eq. (4). Different parameters for the MLPs for different embeddings are possible. After these two decoupled aggregations, we concatenate the embeddings to obtain the final node embedding as a $n \times (2d)$ matrix $\mathbf{Z}_{\mathcal{V}} = \text{CONCAT}\,(\mathbf{Z}_{\mathcal{V}}^s, \mathbf{Z}_{\mathcal{V}}^t)$. The embedding vector $\mathbf{z}_i$ for a node $v_i$ is the $i^{\text{th}}$ row of $\mathbf{Z}_{\mathcal{V}}$, $\mathbf{z}_i := (\mathbf{Z}_{\mathcal{V}})_{(i,:)} \in \mathbb{R}^{2d}$. An efficient implementation of DIMPA is shown in Algorithm 1 in SI B.8.

After obtaining the embedding matrix $\mathbf{Z}_{\mathcal{V}}$, we apply a linear layer (an affine transformation) to $\mathbf{Z}_{\mathcal{V}}$, so that the resulting matrix has $K$ columns. Next, we apply the unit *softmax* function to the rows and obtain the assignment probability matrix $\mathbf{P}$. Figure 2 gives an overview of this implementation.

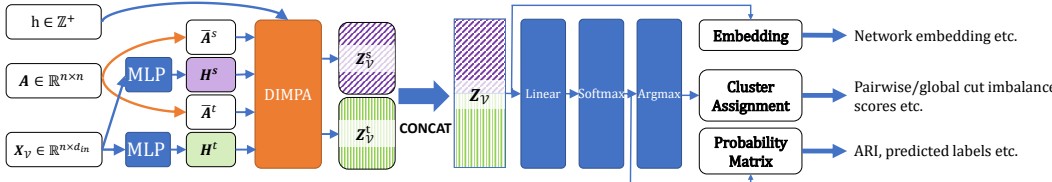

Figure 2: DIGRAC overview: from feature matrix $\mathbf{X}_{\mathcal{V}}$ and adjacency matrix $\mathbf{A}$, we first compute the row-normalized adjacency matrices $\overline{\mathbf{A}}^s$ and $\overline{\mathbf{A}}^t$. Then, we apply two separate MLPs on $\mathbf{X}_{\mathcal{V}}$, to obtain hidden representations $\mathbf{H}^s$ and $\mathbf{H}^t$. Next, we compute their decoupled embeddings using Eq. (4), and its equivalent for target embeddings. The concatenated decoupled embeddings are the final embeddings. For node clustering tasks, we add a linear layer followed by a unit *softmax* to obtain the probability matrix $\mathbf{P}$. Applying *argmax* on each row of $\mathbf{P}$ yields node cluster assignments.

## 4 EXPERIMENTS

In this section, we describe the synthetic and real-world data sets used in this study, and illustrate the efficacy of our method. When ground truth is available, performance is measured by the Adjusted Rand Index (ARI) (Hubert & Arabie, 1985) for node clustering, and by accuracy for node classification and link direction prediction. Normalized Mutual Information (NMI) results give almost the same ranking for the best-performing methods as the ARI, with an average Kendall tau value of 84.2% and standard deviation 25.1%, for pairwise ranking comparison, on the methods compared in our experiments. We do not focus on NMI in the main text due to its shortcomings (Liu et al., 2019).

Clustering tasks will have different ground truths, depending on the pattern they are trying to detect. Many network clustering methods focus on detecting relatively dense clusters, and try to optimize classical network clustering measures, such as directed modularity or partition density. Ground truth for these clustering algorithms then relates to relatively densely connected subgroups in the data. DIGRAC is a novel method that addresses a novel task, namely that of detecting flow imbalances. To the best of our knowledge, real data sets with ground-truth flow imbalances are not available to date, and hence we introduce normalized imbalance scores to evaluate clustering performance based on flow imbalance. As ARI and NMI require ground-truth labels, they thus cannot be applied to the available real-world data sets. To address this shortcoming, for the real-world data sets, in Table 1, we include two performance measures which we introduce in the paper, and the SI contains an additional 12 performance measures. Implementation details are provided in SI B.

In our experiments, we compare DIGRAC against the most recent related methods from the literature for clustering digraphs. The **11** methods are ● (1) InfoMap (Rosvall & Bergstrom, 2008), ● (2) Bibliometric and ● (3) Degree-discounted introduced in (Satuluri & Parthasarathy, 2011), ● (4) DI_SIM (Rohe et al., 2016), ● (5) Herm and ● (6) Herm_sym introduced in Cucuringu et al. (2020a), ● (7) MagNet (Zhang et al., 2021b), ● (8) DGCN (Tong et al., 2020b), ● (9) three variants of DiGCN (Tong et al., 2020a). The abbreviations of these methods, when reported in the numerical experiments, are Bi_sym, DD_sym, DISG_LR, Herm, Herm_sym, MagNet, DGCN,

DiGCN, DiGCN_app, DiGCN_ib, respectively. DGCN is the least efficient method in terms of speed and space complexity, followed by DiGCN_ib which involves the so-called *inception blocks* (hence the suffix *ib*). DiGCN denotes the method without using approximate Laplacian based on personalized PageRank, while DiGCN_app and DiGCN_ib use this approximation. We use the same hyperparameter settings stated in these papers. Methods (7), (8), (9) are trained with 80% nodes under label supervision while all the other methods are trained without label supervision. DIGRAC further restricts itself to be trained on the subgraph induced by only the training nodes. For MagNet, we use $q = 0.25$ for the phase matrix, which is suggested to capture directionality the most effectively.

## 4.1 DATA SETS

**Synthetic data: Directed Stochastic Block Models**   A standard directed stochastic blockmodel (DSBM) is often used to represent a network cluster structure, see for example (Malliaros & Vazirgiannis, 2013). Its parameters are the number $K$ of clusters and the edge probabilities; given the cluster assignment of the nodes, the edge indicators are independent. The DSBMs used in our experiments also depend on a meta-graph adjacency matrix $\mathbf{F} = (\mathbf{F}_{k,l})_{k,l=0,\ldots,K-1}$ and a *filled* version of it, $\tilde{\mathbf{F}} = (\tilde{\mathbf{F}}_{k,l})_{k,l=0,\ldots,K-1}$, and on a noise level parameters $\eta \leq 0.5$. The meta-graph adjacency matrix $\mathbf{F}$ is generated from the given meta-graph structure, called $\mathcal{M}$. To include an ambient background, the filled meta-graph adjacency matrix $\tilde{\mathbf{F}}$ replaces every zero in $\mathbf{F}$ that is not part of the imbalance structure by 0.5. The filled meta-graph thus creates a number of *ambient nodes* which correspond to entries which are not part of $\mathcal{M}$ and thus are not part of a meaningful cluster; this set of *ambient nodes* is also called the *ambient cluster*. First, we provide examples of structures of $\mathbf{F}$ without any ambient nodes, where $\mathbb{1}$ denotes the indicator function.
- (1) "*cycle*": $\mathbf{F}_{k,l} = (1-\eta)\mathbb{1}(l = ((k+1) \mod K)) + \eta\mathbb{1}(l = ((k-1) \mod K)) + \frac{1}{2}\mathbb{1}(l = k)$.
- (2) "*path*": $\mathbf{F}_{k,l} = (1-\eta)\mathbb{1}(l = k+1) + \eta\mathbb{1}(l = k-1) + \frac{1}{2}\mathbb{1}(l = k)$.
- (3) "*complete*": assign diagonal entries $\frac{1}{2}$. For each pair $(k, l)$ with $k < l$, let $\mathbf{F}_{k,l}$ be $\eta$ and $1 - \eta$ with equal probability, then assign $\mathbf{F}_{l,k} = 1 - \mathbf{F}_{k,l}$.
- (4) "*star*", following (Elliott et al., 2019): select the center node as $\omega = \lfloor \frac{K-1}{2} \rfloor$ and set $\mathbf{F}_{k,l} = (1-\eta)\mathbb{1}(k = \omega, l \text{ odd}) + \eta\mathbb{1}(k = \omega, l \text{ even}) + (1-\eta)\mathbb{1}(l = \omega, k \text{ odd}) + \eta\mathbb{1}(l = \omega, k \text{ even})$.

When ambient nodes are present, the construction involves two steps, with the first step the same as the above, but with the following changes: For "*cycle*" meta-graph structure, $\mathbf{F}_{k,l} = (1-\eta)\mathbb{1}(l = ((k+1) \mod (K-1))) + \eta\mathbb{1}(l = ((k-1) \mod (K-1))) + 0.5\,\mathbb{1}(l = k)$. The second step is to assign 0 (0.5, resp.) to the last row and the last column of $\mathbf{F}$ ($\tilde{\mathbf{F}}$, resp.). Figures 1(c-d) display a "*cycle*" meta-graph structure with ambient nodes (in cluster 4). The majority (75%) of edges flow in the form $0 \rightarrow 1 \rightarrow 2 \rightarrow 3 \rightarrow 0$, while 25% flow from the opposite direction. Figure 1(d) illustrate the meta-graph adjacency matrix corresponding to $\mathbf{F}$ shown in Figure 1(c).

In our experiments, we choose the number of clusters, the (approximate) ratio, $\rho$, between the largest and the smallest cluster size, and the number, $n$, of nodes. To tackle the hardest clustering task, all pairs of nodes within a cluster and all pairs of nodes between clusters have the same edge probability, $p$. Note that for $\mathcal{M} =$"*cycle*", even the expected in-degree and out-degree of all nodes are identical. Our DSBM, which we denote by DSBM $(\mathcal{M}, \mathbb{1}(\text{ambient}), n, K, p, \rho, \eta)$, is built similarly to (Cucuringu et al., 2020a) but with possibly unequal cluster sizes, with more details in SI B.3. For each node $v_i \in \mathcal{C}_k$, and each node $v_j \in \mathcal{C}_l$, independently sample an edge from node $v_i$ to node $v_j$ with probability $p \cdot \tilde{\mathbf{F}}_{k,l}$. The parameter settings in our experiments are $p \in \{0.001, 0.01, 0.02, 0.1\}$, $\rho \in \{1, 1.5\}$, $K \in \{3, 5, 10\}$, $\mathbb{1}(\text{ambient}) \in \{\text{T, F}\}$ (True and False), $n \in \{1000, 5000, 30000\}$, and we also vary the direction flip probability $\eta$ from 0 to 0.45, with a 0.05 step size.

**Real-world data**   We perform experiments on five real-world digraph data sets, namely *Telegram* (Bovet & Grindrod, 2020), *Blog* (Adamic & Glance, 2005), *Migration* (Perry, 2003), *WikiTalk* (Leskovec et al., 2010), and *Lead-Lag* (Bennett et al., 2021), with details in SI B.3. We set the number of clusters $K$ to be 4, 2, 10, 10, 10, respectively, and values of $\beta$ to be 5, 1, 9, 10, 3, respectively. Note that *Lead-Lag* comprises of 19 separate networks constructed from yearly financial time series.

## 4.2 EXPERIMENTAL RESULTS

**Training set-up and hyperparameter selection**   As training setup, we use 10% of all nodes from each cluster as test nodes, 10% as validation nodes to select the model, and the remaining 80% as training nodes. In each setting, unless otherwise stated, we carry out 10 experiments with different

data splits. Error bars are given by one standard error. Without node attributes, the matrix $\mathbf{X}_\mathcal{V}$ for DIGRAC is taken as the stacked eigenvectors corresponding to the largest $K$ eigenvalues of the random-walk symmetrized Hermitian matrix used in the comparison method Herm_rw. The imbalance loss function acts on the subgraph induced by the training nodes. Hyperparameters are selected via greedy search, with details in SI B.4. Figure 3(a) compares DIGRAC on DSBM with $n = 1000$ nodes, $K = 5$ clusters, $\rho = 1, p = 0.02$ without ambient nodes, under different hyperparameter settings. Here hop $h = 2$ should be chosen to reduce complexity, as increasing $h$ does not lead to increased performance. Therefore, our default setting is hop $h = 2, d = 32, \tau = 0.5$.

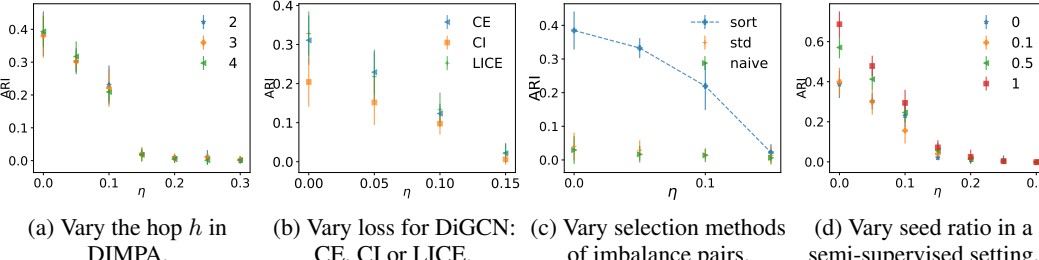

(a) Vary the hop $h$ in DIMPA.

(b) Vary loss for DiGCN: CE, CI or LICE.

(c) Vary selection methods of imbalance pairs.

(d) Vary seed ratio in a semi-supervised setting.

Figure 3: Hyperparemter analysis (a) and ablation study (b-d) on a "*cycle*" DSBM with $n = 1000$ nodes, $K = 5$ clusters, $\rho = 1$, and $p = 0.02$, without ambient nodes.

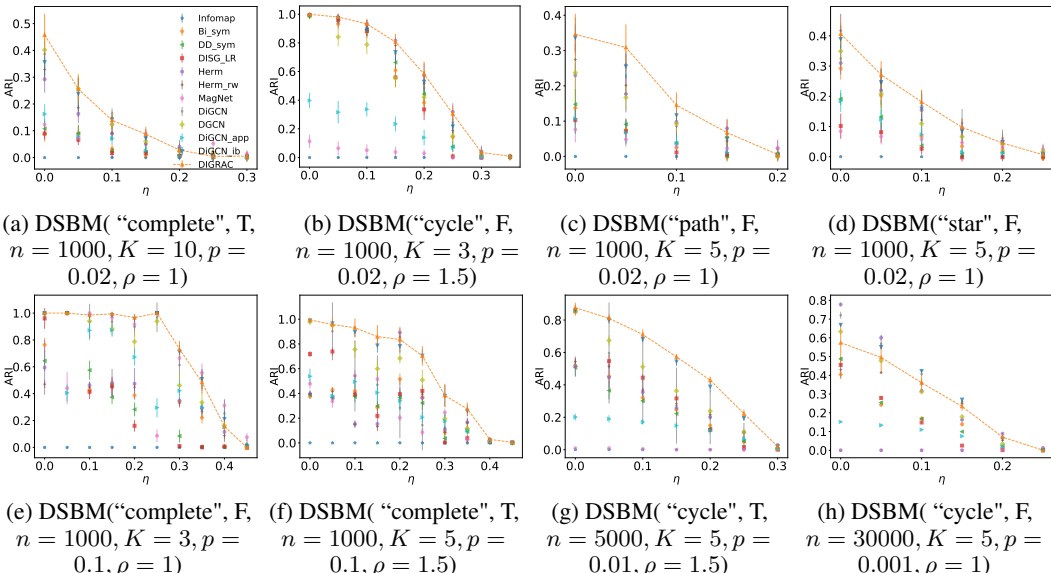

(a) DSBM( "complete", T, $n = 1000, K = 10, p = 0.02, \rho = 1$)

(b) DSBM("cycle", F, $n = 1000, K = 3, p = 0.02, \rho = 1.5$)

(c) DSBM("path", F, $n = 1000, K = 5, p = 0.02, \rho = 1$)

(d) DSBM("star", F, $n = 1000, K = 5, p = 0.02, \rho = 1$)

(e) DSBM("complete", F, $n = 1000, K = 3, p = 0.1, \rho = 1$)

(f) DSBM( "complete", T, $n = 1000, K = 5, p = 0.1, \rho = 1.5$)

(g) DSBM( "cycle", T, $n = 5000, K = 5, p = 0.01, \rho = 1.5$)

(h) DSBM( "cycle", F, $n = 30000, K = 5, p = 0.001, \rho = 1$)

Figure 4: Test ARI comparison on synthetic data. Dashed lines highlight DIGRAC's performance. Error bars are given by one standard error.

**Node clustering results on synthetic data** Figure 4 compares the numerical performance of DIGRAC with other methods on synthetic data. For this figure we generate 5 DSBM networks under each parameter setting and use 10 different data splits for each network, then average over the 50 runs. Error bars are given by one standard error. Additional implementation details are available in SI B.

We conclude that DIGRAC attains state-of-the-art results on a wide range of network densities and noise levels, on different network sizes, and with different meta-graph structures, whether or not there exist ambient nodes. DIGRAC outperforms its competitors by a large margin especially when there exist ambient nodes, which validates its strength of extracting the main directional signal planted in a larger graph. Additional results are reported in SI C, from which similar conclusions can be drawn.

**Node clustering results on real-world data** For our real-world data sets, the node in- and out-degrees may not be identical across clusters. Moreover, as these data sets do not contain node attributes, DIGRAC considers the eigenvectors corresponding to the largest $K$ eigenvectors of the Hermitian matrix from (Cucuringu et al., 2020a) to construct an input feature matrix. Table 1 reveals

Table 1: Performance comparison on real-world data sets. The best is marked in **bold red** and the second best is marked in underline blue. The objectives are defined in Section 3.1.

| Metric | Data set | InfoMap | Bi_sym | DD_sym | DISG_LR | Herm | Herm_rw | DIGRAC |
|---|---|---|---|---|---|---|---|---|
| $\mathcal{O}_{\text{vol\_sum}}^{\text{sort}}$ | *Telegram* | 0.04±0.00 | 0.21±0.0 | 0.21±0.0 | 0.21±0.01 | 0.2±0.01 | 0.14±0.0 | **0.32±0.01** |
| | *Blog* | 0.07±0.00 | 0.07±0.0 | 0.0±0.0 | 0.05±0.0 | 0.37±0.0 | 0.0±0.0 | **0.44±0.0** |
| | *Migration* | N/A | 0.03±0.00 | 0.01±0.00 | 0.02±0.00 | 0.04±0.00 | 0.02±0.00 | **0.05±0.00** |
| | *WikiTalk* | N/A | N/A | N/A | 0.18±0.03 | 0.15±0.02 | 0.0±0.0 | **0.24±0.05** |
| | *Lead-Lag* | N/A | 0.07±0.01 | 0.07±0.01 | 0.07±0.01 | 0.07±0.02 | 0.07±0.02 | **0.15±0.03** |
| $\mathcal{O}_{\text{vol\_sum}}^{\text{naive}}$ | *Telegram* | 0.01±0.00 | 0.26±0.0 | 0.26±0.0 | 0.26±0.01 | 0.25±0.02 | 0.23±0.0 | **0.27±0.01** |
| | *Blog* | 0.00±0.00 | 0.07±0.0 | 0.0±0.0 | 0.05±0.0 | 0.37±0.0 | 0.0±0.0 | **0.44±0.0** |
| | *Migration* | N/A | 0.01±0.00 | 0.01±0.00 | 0.01±0.00 | 0.02±0.00 | 0.01±0.00 | **0.04±0.01** |
| | *WikiTalk* | N/A | N/A | N/A | 0.1±0.02 | 0.04±0.0 | 0.0±0.0 | **0.12±0.01** |
| | *Lead-Lag* | N/A | 0.30±0.06 | 0.28±0.06 | 0.27±0.06 | 0.29±0.05 | 0.29±0.05 | **0.32±0.11** |

that DIGRAC provides competitive global imbalance scores in both objectives discussed and across all real-world data sets, and outperforms all other methods in all instances, on the four data sets and two objective functions. The N/A entries for *WikiTalk* are caused by memory error, and the N/A entries for InfoMap on *Migration* and *Lead-Lag* are due to its prediction of only one single cluster. *Lead-Lag* results in each year are averaged over ten runs, while the mean and standard deviation values are calculated with respect to the 19 years. The experiments indicate that edge directionality contains an important signal that DIGRAC is able to capture. A comprehensive numerical comparison is available in SI D, revealing similar conclusions.

## 4.3 ABLATION STUDY

Figure 3(b) compares the performance of DiGCN replacing the loss function by $\mathcal{L}_{\text{vol\_sum}}^{\text{sort}}$ from Eq. (3), indicated by "CI", or LICE, sum of supervised and self-supervised loss, on a DSBM("*cycle*", T, $n = 1000, K = 5, p = 0.02, \rho = 1$) model. We find that replacing the supervised loss function with $\mathcal{L}_{\text{vol\_sum}}^{\text{sort}}$ leads to comparable results, and that adding $\mathcal{L}_{\text{vol\_sum}}^{\text{sort}}$ to the loss could be beneficial. Figure 3(c) compares the test ARI performance using three variants of loss functions on the same digraph. The current choice "*sort*" performs best among these variants, indicating a benefit in only considering top pairs of individual imbalance scores. More details on loss functions, comparison with other variants, and evaluation on additional metrics are discussed in SI A, with similar conclusions. As illustrated in Figure 3(d), again on the same digraph, we also experiment on adding seeds, with the seed ratio defined as the ratio of the number of seed nodes to the number of training nodes. A supervised loss, following (Tian et al., 2019), is then applied to these seeds; SI B.5 contains additional details. In conclusion, seed nodes with a supervised loss function enhance performance, and we infer that our model can further boost its performance when additional label information is available.

## 5 CONCLUSION AND FUTURE WORK

DIGRAC provides an end-to-end pipeline to create node embeddings and perform directed clustering, with or without available additional node features or cluster labels. Further work will include additional experiments in the semi-supervised setting, when there exist seed nodes with known cluster labels, or when additional information is available in the form of *must-link* and *cannot-link* constraints, popular in the *constrained clustering* literature (Basu et al., 2008; Cucuringu et al., 2016). Another future direction pertains to extending our framework to also detect the number of clusters (Riolo et al., 2017; Chen & Baker, 2021), instead of specifying it a-priori, as this is typically not available in real-world applications. The current framework requires additional preliminary analysis on how many pairwise imbalance scores to consider, such as by inspecting the (initial or fitted) meta-graph adjacency matrix. It would be interesting to build a more powerful framework that can automatically detect the value $\beta$ used in the current model, to select the subset of influential pairs of imbalances.

Further research directions will address the performance in the sparse regime, where spectral methods are known to underperform, and various regularizations have been proven to be effective both on the theoretical and experimental fronts; for example, see regularization in the sparse regime for the undirected settings (Chaudhuri et al., 2012; Amini et al., 2013; Cucuringu et al., 2020b). Finally, adapting our pipeline for directed clustering in extremely large networks, possibly combined with sampling methods or mini-batch (Hamilton et al., 2017), is a direction worth exploring, rendering DIGRAC applicable to large scale industrial applications.

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

## A    LOSS AND OBJECTIVES

### A.1    ADDITIONAL DETAILS ON PROBABILISTIC CUT AND VOLUME

Recall that the **probabilistic cut** from cluster $\mathcal{C}_k$ to $\mathcal{C}_l$ is defined as

$$W(\mathcal{C}_k, \mathcal{C}_l) = \sum_{i,j \in \{1,\dots,n\}} \mathbf{A}_{i,j} \cdot \mathbf{P}_{i,k} \cdot \mathbf{P}_{j,l} = (\mathbf{P}_{(:,k)})^T \mathbf{A} \mathbf{P}_{(:,l)},$$

where $\mathbf{P}_{(:,k)}, \mathbf{P}_{(:,l)}$ denote the $k^{\text{th}}$ and $l^{\text{th}}$ columns of the assignment probability matrix $\mathbf{P}$, respectively. The **imbalance flow** between clusters $\mathcal{C}_k$ and $\mathcal{C}_l$ is defined as

$$|W(\mathcal{C}_k, \mathcal{C}_l) - W(\mathcal{C}_l, \mathcal{C}_k)|,$$

for $k, l \in \{0, \dots, K-1\}$. The loss functions proposed in the main paper can be understood in terms of a probabilistic notion of degrees, as follows. We define the probabilistic out-degree of node $v_i$ with respect to cluster $k$ by $\tilde{d}_{i,k}^{(\text{out})} = \sum_{j=1}^{n} \mathbf{A}_{i,j} \cdot \mathbf{P}_{j,k} = (\mathbf{A}\mathbf{P}_{(:,k)})_i$, where subscript $i$ refers to the $i^{\text{th}}$ entry of the vector $\mathbf{A}\mathbf{P}_{(:,k)}$. Similarly, we define the probabilistic in-degree of node $v_i$ with respect to cluster $k$ by $\tilde{d}_{i,k}^{(\text{in})} = (\mathbf{A}^T \mathbf{P}_{(:,k)})_i$, where $\mathbf{A}^T$ is the transpose of $\mathbf{A}$. The **probabilistic degree** of node $v_i$ with respect to cluster $k$ is $\tilde{d}_{i,k} = \tilde{d}_{i,k}^{(\text{in})} + \tilde{d}_{i,k}^{(\text{out})} = ((\mathbf{A}^T + \mathbf{A})\mathbf{P}_{(:,k)})_i = \sum_{j=1}^{n}(\mathbf{A}_{i,j} + \mathbf{A}_{j,i}) \cdot \mathbf{P}_{j,k}$.

For comparisons and ease of interpretation, it is advantageous to normalize the imbalance flow between clusters; for this purpose, we introduce the probabilistic volume of a cluster, as follows. The *probabilistic out-volume* for cluster $\mathcal{C}_k$ is defined as $VOL^{(\text{out})}(\mathcal{C}_k) = \sum_{i,j} \mathbf{A}_{j,i} \cdot \mathbf{P}_{j,k}$, and the *probabilistic in-volume* for cluster $\mathcal{C}_k$ is defined as $VOL^{(\text{in})}(\mathcal{C}_k)(\mathbf{A}^T \mathbf{P}_{(:,k)})_i$, where $\mathbf{A}^T$ is the transpose of $\mathbf{A}$. These volumes can be viewed as sum of probabilistic out-degrees and in-degrees, respectively; for example, $VOL^{(\text{in})}(\mathcal{C}_k) = \sum_{i=1}^{n} \tilde{d}_{i,k}^{(\text{in})}$. Then, it holds true that

$$VOL^{(\text{out})}(\mathcal{C}_k) = \sum_{i,j} \mathbf{A}_{i,j} \cdot \mathbf{P}_{i,k} \geq \sum_{i,j} \mathbf{A}_{i,j} \cdot \mathbf{P}_{i,k} \cdot \mathbf{P}_{j,l} = W(\mathcal{C}_k, \mathcal{C}_l), \tag{5}$$

since entries in $\mathbf{P}$ are probabilities, which are in $[0, 1]$, and all entries of $\mathbf{A}$ are nonnegative. Similarly, $VOL^{(\text{in})}(\mathcal{C}_k) \geq W(\mathcal{C}_l, \mathcal{C}_k)$.

The **probabilistic volume** for cluster $\mathcal{C}_k$ is defined as

$$VOL(\mathcal{C}_k) = VOL^{(\text{out})}(\mathcal{C}_k) + VOL^{(\text{in})}(\mathcal{C}_k) = \sum_{i,j}(\mathbf{A}_{i,j} + \mathbf{A}_{j,i}) \cdot \mathbf{P}_{j,k}.$$

Then, it holds true that $VOL(\mathcal{C}_k) \geq W(\mathcal{C}_k, \mathcal{C}_l)$ for all $l \in \{0, \dots, K-1\}$ and

$$\min(VOL(\mathcal{C}_k), VOL(\mathcal{C}_l)) \geq \max(W(\mathcal{C}_k, \mathcal{C}_l), W(\mathcal{C}_l, \mathcal{C}_k)) \geq |W(\mathcal{C}_k, \mathcal{C}_l) - W(\mathcal{C}_l, \mathcal{C}_k)|. \tag{6}$$

When there exists a strong imbalance, then $|W(\mathcal{C}_k, \mathcal{C}_l) - W(\mathcal{C}_l, \mathcal{C}_k)| \approx \max(W(\mathcal{C}_k, \mathcal{C}_l), W(\mathcal{C}_l, \mathcal{C}_k))$. As an extreme case, if $\mathbf{P}_{j,l} = 1$ for all nonnegative terms in the summations in Eq. (5), and $VOL^{(\text{in})}(\mathcal{C}_k) = 0$, then $|W(\mathcal{C}_k, \mathcal{C}_l) - W(\mathcal{C}_l, \mathcal{C}_k)| = VOL(\mathcal{C}_k)$.

### A.2    VARIANTS OF NORMALIZATION

Recall that the imbalance term involved in most of our experiments, named $\text{CI}^{\text{vol\_sum}}$, is defined as

$$\text{CI}^{\text{vol\_sum}}(k, l) = 2\frac{|W(\mathcal{C}_k, \mathcal{C}_l) - W(\mathcal{C}_l, \mathcal{C}_k)|}{VOL(\mathcal{C}_k) + VOL(\mathcal{C}_l)} \in [0, 1]. \tag{7}$$

An alternative, which does not take volumes into account, is given by

$$\text{CI}^{\text{plain}}(k, l) = \left| \frac{W(\mathcal{C}_k, \mathcal{C}_l) - W(\mathcal{C}_l, \mathcal{C}_k)}{W(\mathcal{C}_k, \mathcal{C}_l) + W(\mathcal{C}_l, \mathcal{C}_k)} \right| = 2\left| \frac{W(\mathcal{C}_k, \mathcal{C}_l)}{W(\mathcal{C}_k, \mathcal{C}_l) - W(\mathcal{C}_l, \mathcal{C}_k)} - \frac{1}{2} \right| \in [0, 1]. \tag{8}$$

We call this cut flow imbalance $\text{CI}^{\text{plain}}$ as it does not penalize extremely unbalanced cluster sizes.

To achieve balanced cluster sizes and still constrain each imbalance term to be in $[0, 1]$, one solution is to multiply the imbalance flow value by the minimum of $VOL(\mathcal{C}_k)$ and $VOL(\mathcal{C}_l)$, and then divide by

$\max_{(k',l')\in\mathcal{T}}(\min(VOL(\mathcal{C}_{k'}),VOL(\mathcal{C}_{l'})))$, where $\mathcal{T} = \{(\mathcal{C}_k,\mathcal{C}_l) : 0 \leq k < l \leq K - 1, k, l \in \mathbb{Z}\}$. The reason for using $\mathcal{T}$ is that $\text{CI}^{\text{plain}}(k,l)$ is symmetric with respect to $k$ and $l$, and $\text{CI}^{\text{plain}}(k,l) = 0$ whenever $k = l$. Note that the maximum of the minimum here equals the second largest volume among clusters. We then obtain $\text{CI}^{\text{vol\_min}}$ as

$$\text{CI}^{\text{vol\_min}}(k,l) = \text{CI}^{\text{plain}}(k,l) \times \frac{\min(VOL(\mathcal{C}_k),VOL(\mathcal{C}_l))}{\max_{(k',l')\in\mathcal{T}}(\min(VOL(\mathcal{C}_{k'}),VOL(\mathcal{C}_{l'})))}. \tag{9}$$

Another potential choice, denoted $\text{CI}^{\text{vol\_max}}$, whose normalization follows from the same reasoning as $\text{CI}^{\text{vol\_sum}}$, is given by

$$\text{CI}^{\text{vol\_max}}(k,l) = \frac{|W(\mathcal{C}_k,\mathcal{C}_l) - W(\mathcal{C}_l,\mathcal{C}_k)|}{\max(VOL(\mathcal{C}_k),VOL(\mathcal{C}_l))} \in [0, 1]. \tag{10}$$

### A.3 Variants of choosing the pairwise imbalance scores

We consider three variants for choosing the cluster pairs.
• (1) The **"sort"** variant picks the largest $\beta$ pairwise cut imbalance values, where $\beta$ is half of the number of nonzero entries in the off-diagonal entries of the meta-graph adjacency matrix $\mathbf{F}$, if the meta-graph is known or can be approximated. For example, when we have a "cycle" meta-graph with three clusters and no ambient nodes, then $\beta = 3$. When we have a "path" meta-graph with three clusters and ambient nodes, then $\beta = 1$.
• (2) The **"naive"** variant considers all possible $\binom{K}{2}$ pairwise cut imbalance values.
• (3) The **"std"** variant only considers pairwise cut imbalance values that are 3 standard deviations away from the imbalance values; the standard deviation is calculated under the null hypothesis that the between-cluster relationship has no direction preference, i.e. $\mathbf{F}_{k,l} = \mathbf{F}_{l,k}$, as follows.

Suppose two clusters have only noisy links between them (no edge in the meta-graph $\mathbf{F}$). Suppose also that the underlying network is fixed in terms of the number of nodes and where edges exist; the only randomness stems from the direction of an edge. Then, for each edge between these two clusters, say, clusters $\mathcal{C}_k$ and $\mathcal{C}_l$, the edge direction is random, i.e. the edge is from $\mathcal{C}_k$ to $\mathcal{C}_l$ with probability 0.5, and $\mathcal{C}_l$ to $\mathcal{C}_k$ with probability 0.5 also. Let $\mathcal{E}^{k,l}$ denote the set of edges between $\mathcal{C}_k$ and $\mathcal{C}_l$ if $\mathcal{E}^{k,l}$ is not empty, then for every edge $e \in \mathcal{E}^{k,l}$, define a Rademacher random variable $X_e$ by

$$X_e = \begin{cases} 1 & \text{if the edge is from } \mathcal{C}_k \text{ to } \mathcal{C}_l, \\ -1, & \text{otherwise.} \end{cases} \tag{11}$$

Then $(X_e + 1)/2 \sim Ber(0.5)$ is a Bernoulli(0.5) random variable with mean $2 \times 0.5 - 1 = 0$ and variance $2^2 \times 0.5 \times (1 - 0.5) = 1$. In the case of unweighted edges, the total number of edges between $\mathcal{C}_k$ and $\mathcal{C}_l$ is $|\mathcal{E}^{k,l}| = W(\mathcal{C}_k,\mathcal{C}_l) + W(\mathcal{C}_l,\mathcal{C}_k)$, and that the sum of $X_e$ terms is $\sum_{e\in\mathcal{E}^{k,l}} X_e = W(\mathcal{C}_k,\mathcal{C}_l) - W(\mathcal{C}_l,\mathcal{C}_k)$. In the case of weighted edges, with symmetric edge weights $w_{i,j} = w_{j,i}$ given and only edge direction random, it holds that $W(\mathcal{C}_k,\mathcal{C}_l) - W(\mathcal{C}_l,\mathcal{C}_k) = \sum_{e\in\mathcal{E}^{k,l}} X_e w_e$.

Let us assume that the edge indicators are independent and that $\sum_{e\in\mathcal{E}^{k,l}} w_e^2 > 0$. Under the null hypothesis that there is no meta-graph edge between $\mathcal{C}_k$ and $\mathcal{C}_l$, the random variable $\frac{\sum_{e\in\mathcal{E}^{k,l}} X_e w_e}{\sqrt{\sum_{e\in\mathcal{E}^{k,l}} w_e^2}}$ has mean 0 and variance 1. Assuming that the weights are bounded above and that $\sum_{e\in\mathcal{E}^{k,l}} w_e^2$ is bounded away from 0 with increasing network size, we can employ the Central Limit Theorem for sums of independent random variables, see for example Theorem 3.4 in Chen et al. (2010). Then, under the null hypothesis, approximately 99.7 % of the observations would fall within 3 standard deviations from 0. While this calculation makes many assumptions and ignores reciprocal edges, the resulting threshold is still a useful guideline for restricting attention to pairwise imbalance values which are very likely to capture a true signal.

### A.4 Selection of the loss function

Table 2 provides naming conventions of all the twelve pairs of variants of objectives and loss functions used in this paper. We select the loss functions for DIGRAC based on two representative models, and compare the performance of different loss functions. We use $d = 32$, hidden units, $h = 2$ hops,

Table 2: Naming conventions for objectives and loss functions

| Selection variant / $CI$ | $CI^{\text{vol\_sum}}$ | $CI^{\text{vol\_min}}$ | $CI^{\text{vol\_max}}$ | $CI^{\text{plain}}$ |
|---|---|---|---|---|
| sort | $\mathcal{O}^{\text{sort}}_{\text{vol\_sum}}, \mathcal{L}^{\text{sort}}_{\text{vol\_sum}}$ | $\mathcal{O}^{\text{sort}}_{\text{vol\_min}}, \mathcal{L}^{\text{sort}}_{\text{vol\_min}}$ | $\mathcal{O}^{\text{sort}}_{\text{vol\_max}}, \mathcal{L}^{\text{sort}}_{\text{vol\_max}}$ | $\mathcal{O}^{\text{sort}}_{\text{plain}}, \mathcal{L}^{\text{sort}}_{\text{plain}}$ |
| std | $\mathcal{O}^{\text{std}}_{\text{vol\_sum}}, \mathcal{L}^{\text{std}}_{\text{vol\_sum}}$ | $\mathcal{O}^{\text{std}}_{\text{vol\_min}}, \mathcal{L}^{\text{std}}_{\text{vol\_min}}$ | $\mathcal{O}^{\text{std}}_{\text{vol\_max}}, \mathcal{L}^{\text{std}}_{\text{vol\_max}}$ | $\mathcal{O}^{\text{std}}_{\text{plain}}, \mathcal{L}^{\text{std}}_{\text{plain}}$ |
| naive | $\mathcal{O}^{\text{naive}}_{\text{vol\_sum}}, \mathcal{L}^{\text{naive}}_{\text{vol\_sum}}$ | $\mathcal{O}^{\text{naive}}_{\text{vol\_min}}, \mathcal{L}^{\text{naive}}_{\text{vol\_min}}$ | $\mathcal{O}^{\text{naive}}_{\text{vol\_max}}, \mathcal{L}^{\text{naive}}_{\text{vol\_max}}$ | $\mathcal{O}^{\text{naive}}_{\text{plain}}, \mathcal{L}^{\text{naive}}_{\text{plain}}$ |

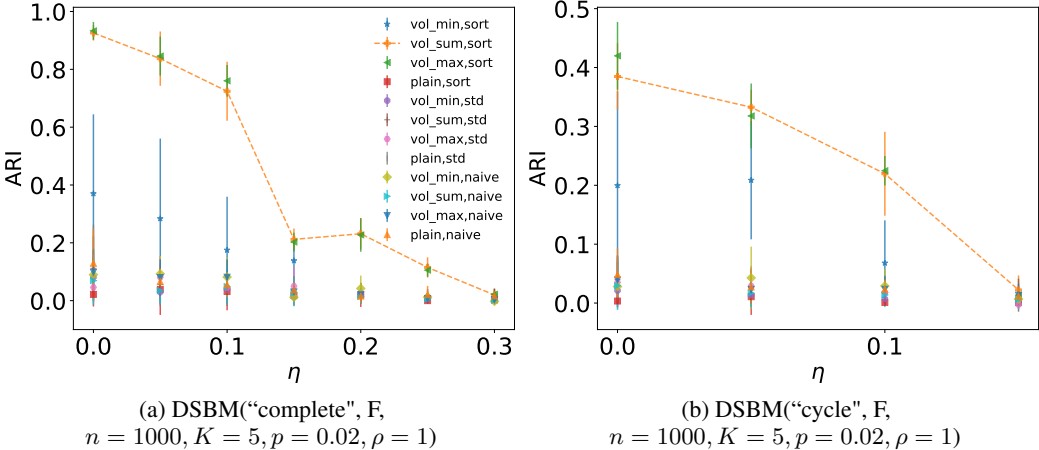

(a) DSBM("complete", F, $n = 1000, K = 5, p = 0.02, \rho = 1$)

(b) DSBM("cycle", F, $n = 1000, K = 5, p = 0.02, \rho = 1$)

Figure 5: ARI comparison of loss functions on DSBM with 1000 nodes, 5 blocks, $\rho = 1, p = 0.02$ without ambient nodes, of cycle (left) and complete (right) meta-graph structures, respectively. The first component of the legend is the choice of pairwise imbalance, and the second component is the variant of selecting pairs. The naming conventions for the abbreviations in the legend are provided in Table 2.

and no seed nodes. Figures 5(a) and 6 compare twelve choices of loss combinations on a DSBM with $n = 1000$ nodes, $K = 5$ blocks, $\rho = 1, p = 0.02$ without ambient nodes, with a complete meta-graph structure. The subscript indicates the choice of pairwise imbalance, and the superscript indicates the variant of selecting pairs. Figures 5(b) and 7 are based on a DSBM with $n = 1000$ nodes, $K = 5$ blocks, $\rho = 1, p = 0.02$ without ambient nodes, with a cycle meta-graph structure.

These figures indicate that the "*sort*" variant generally provides the best test ARI performance and the best overall global imbalance scores, among which using normalizations $\text{CI}^{\text{vol\_sum}}$ and $\text{CI}^{\text{vol\_max}}$ perform the best. $\mathcal{L}^{\text{sort}}_{\text{vol\_min}}$ appears to behave worse than $\mathcal{L}^{\text{sort}}_{\text{vol\_sum}}$ and $\mathcal{L}^{\text{sort}}_{\text{vol\_max}}$, even when using the "sort" variant to select pairwise imbalance scores. One possible explanation is that $\mathcal{L}^{\text{sort}}_{\text{vol\_min}}$ does not penalize extreme volume sizes, and that it takes minimum as well as maximum which, as functions of the data, are not as smooth as taking a summation. Throughout our experiments in the main text, we hence use the loss function $\mathcal{L}^{\text{sort}}_{\text{vol\_sum}}$.

# B   Implementation details

## B.1   Code

To fully reproduce our results, anonymized code and preprocessed data are available at `https://anonymous.4open.science/r/1b728e97-cc2b-4e6a-98ea-37668813536c`.

## B.2   Hardware

Experiments were conducted on a compute node with 8 Nvidia RTX 8000, 48 Intel Xeon Silver 4116 CPUs and 1000GB RAM, a compute node with 4 NVIDIA GeForce RTX 2080, 32 Intel Xeon

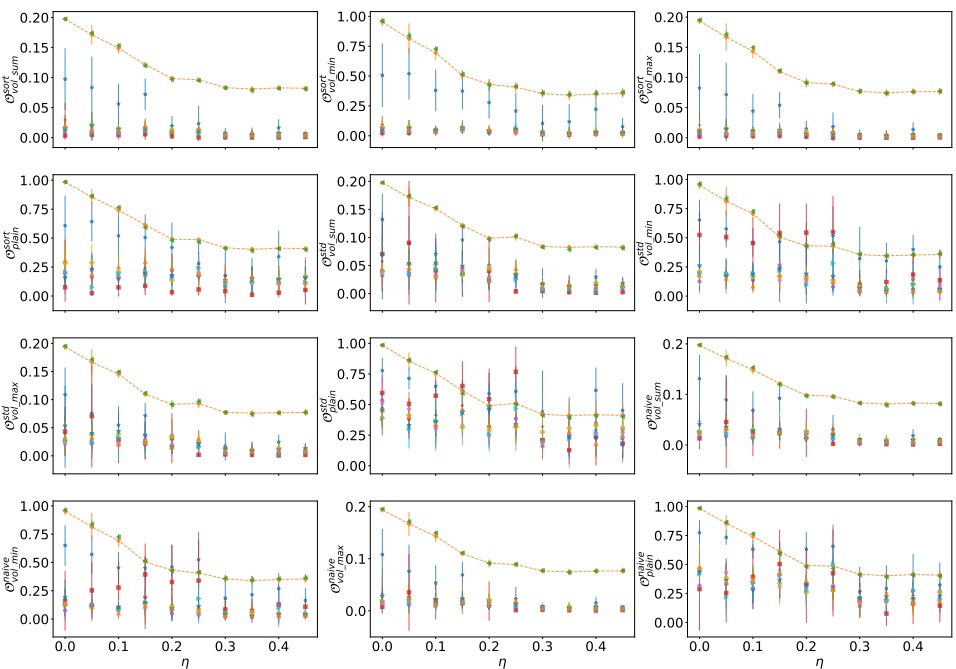

Figure 6: Imbalance scores comparison of loss functions on DSBM with 1000 nodes, 5 blocks, $\rho = 1, p = 0.02$ without ambient nodes, of the **complete meta-graph** structure. The legend is the same as Figure 5(a).

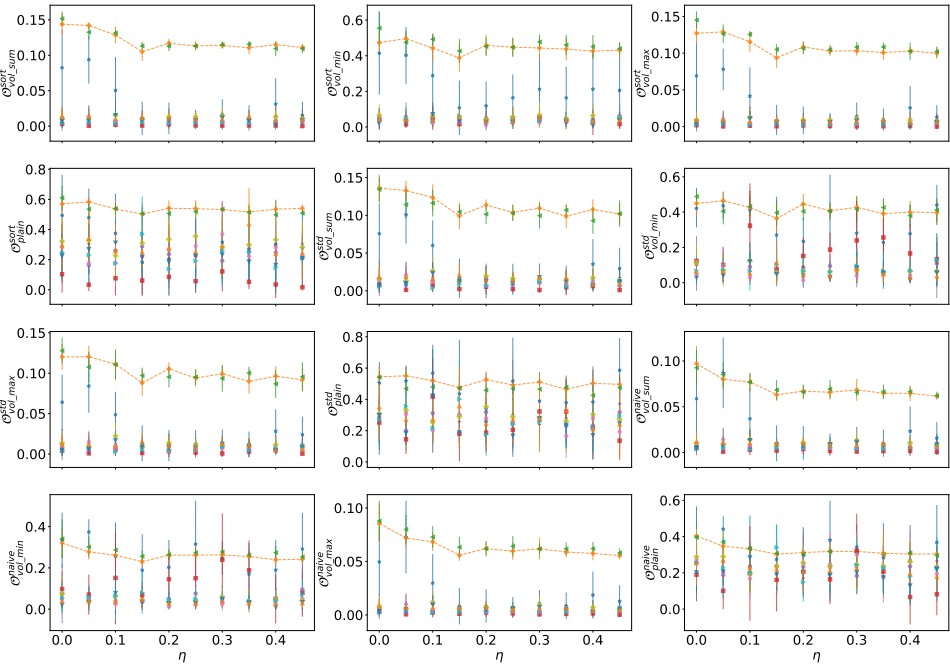

Figure 7: Imbalance scores comparison of loss functions on DSBM with 1000 nodes, 5 blocks, $\rho = 1, p = 0.02$ without ambient nodes, of the **cyclic meta-graph** structure. The legend is the same as Figure 5(a).

E5-2690 v3 CPUs and 64GB RAM, a compute node with 2 NVIDIA Tesla K80, 16 Intel Xeon E5-2690 CPUs and 252GB RAM, and an Intel 2.90GHz i7-10700 processor with 8 cores and 16 threads.

With this setup, all experiments for spectral methods, MagNet, DiGCN and DIGRAC can be completed within two days, including repeated experiments, to obtain averages over multiple runs. DGCN and DiGCN_ib have much longer run time (especially DGCN, which is space-consuming, and we cannot run many experiments in parallel), with a total of three days for both of them to finish. The slow speed stems from the competitor methods; some of the other GNN methods take a long time to run. Table 1 in the main text shows N/A values for Bi_sym and for DD_sym exactly for this reason. Empirically, DIGRAC is the fastest among all GNN methods to which it is compared.

### B.3 DATA

The results comparing DIGRAC with other methods on synthetic data are averaged over 50 runs, five synthetic networks under the same setting, each with 10 different data splits. For synthetic data, 10% of all nodes are selected as test nodes for each cluster (the actual number is the ceiling of the total number of nodes times 0.1, to avoid falling below 10% of test nodes), 10% are selected as validation nodes (for model selection and early-stopping; again, we consider the ceiling for the actual number), while the remaining roughly 80% are selected as training nodes (the actual number can never be higher than 80% due to using the ceiling for both the test and validation splits). To further clarify the training setup, we use 0% of the labels in training. As DIGRAC is a self-supervised method in principle we could use all nodes for training. However for a fair comparison with other GNN methods we use only 80% of the nodes for training. For supervised methods our split of 80% - 10% - 10% is a standard split. For the non-GNN methods, all nodes are used for training.

For both synthetic and real-world data sets, we extract the largest weakly connected component for experiments, as our framework could be applied to different weakly connected components, if the digraph is disconnected. Isolated nodes do not include any imbalance information. As customary in community detection, they are often omitted in real networks. When "ground-truth" is given, test results are averaged over 10 different data splits on one network. When no labels are available, results are averaged over 10 different data splits.

Averaged results are reported with error bars representing one standard deviation in the figures, and plus/minus one standard deviation in the tables.

Our synthetic data, DSBM, which we denote by DSBM $(\mathcal{M}, \mathbb{1}(\text{ambient}), n, K, p, \rho, \eta)$, is built similarly to (Cucuringu et al., 2020a) but with possibly unequal cluster sizes: • (1) Assign cluster sizes $n_0 \leq n_1 \leq \cdots \leq n_{K-1}$ with size ratio $\rho \geq 1$ , as follows. If $\rho = 1$ then the first $K-1$ clusters have the same size $\lfloor n/K \rfloor$ and the last cluster has size $n - (K-1)\lfloor n/K \rfloor$. If $\rho > 1$, we set $\rho_0 = \rho^{\frac{1}{K-1}}$. Solving $\sum_{i=0}^{K-1} \rho_0^i n_0 = n$ and taking integer value gives $n_0 = \lfloor n(1-\rho_0)/(1-\rho_0^K) \rfloor$ . Further, set $n_i = \lfloor \rho_0 n_{i-1} \rfloor$, for $i = 1, \cdots, K-2$ if $K \geq 3$, and $n_{K-1} = n - \sum_{i=0}^{K-2} n_i$. Then the ratio of the size of the largest to the smallest cluster is approximately $\rho_0^{K-1} = \rho$. • (2) Assign each node randomly to one of $K$ clusters, so that each cluster has the allocated size. • (3) For node $v_i, v_j \in \mathcal{C}_k$, independently sample an edge from node $v_i$ to node $v_j$ with probability $p \cdot \tilde{\mathbf{F}}_{k,k}$. • (4) For each pair of different clusters $\mathcal{C}_k, \mathcal{C}_l$ with $k \neq l$, for each node $v_i \in \mathcal{C}_k$, and each node $v_j \in \mathcal{C}_l$, independently sample an edge from node $v_i$ to node $v_j$ with probability $p \cdot \tilde{\mathbf{F}}_{k,l}$.

For real-world data sets, we choose the number $K$ of clusters in the meta-graph and the number $\beta$ of edges between clusters in the meta-graph as follows. As they are needed as input for DIGRAC, we resort to Herm_rw (Cucuringu et al., 2020a) as an initial view of the network clustering. When a suitable meta-graph is suggested in a previous publication, then we use that choice. Otherwise, the number $K$ of clusters is determined using the clustering from Herm_rw. First, we pick a range of $K$, and for each $K$, we calculate the global imbalance scores and plot the predicted meta-graph flow matrix $\mathbf{F}'$ based on the clustering from Herm_rw. Its entries are defined as

$$\mathbf{F}'(k,l) = \mathbb{1}(W(\mathcal{C}_k, \mathcal{C}_l) + W(\mathcal{C}_l, \mathcal{C}_k) > 0) \times \frac{W(\mathcal{C}_k, \mathcal{C}_l)}{W(\mathcal{C}_k, \mathcal{C}_l) + W(\mathcal{C}_l, \mathcal{C}_k)}. \tag{12}$$

These entries can be viewed as predicted probabilities of edge directions. Then, we choose $K$ from this range so that the predicted meta-graph flow matrix has the highest imbalance scores and strong imbalance in the predicted meta-graph flow matrix.

The choice of $\beta$ is as follows. We plot the ranked pairs of CI$^{\text{plain}}$ values from Herm_rw and select the $\beta$ which is at least as large as $K - 2$, to allow the meta-graph to be connected, and which corresponds to a large drop in the plot.

Here we provide a brief description for each of the data sets; Table 3 gives the number, $n$, of nodes, the number, $|\mathcal{E}|$, of directed edges, the number $|\mathcal{E}^r|$, of reciprocal edges (self-loops are counted once and for $u \neq v$, a reciprocal edge $u \to v, v \to u$ is counted twice) as well as their percentage among all edges, for the real-world networks, illustrating the variability in network size and density (defined as $|\mathcal{E}|/[n(n-1)]$).
• *Telegram* (Bovet & Grindrod, 2020) is a pairwise influence network between $n = 245$ Telegram channels with $|\mathcal{E}| = 8,912$ directed edges. It is found in (Bovet & Grindrod, 2020) that this network reveals a core-periphery structure in the sense of (Elliott et al., 2020). Following (Bovet & Grindrod, 2020) we assume $K = 4$ clusters, and the core-periphery structures gives $\beta = 5$.
• *Blog* (Adamic & Glance, 2005) records $|\mathcal{E}| = 19,024$ directed edges between $n = 1,212$ political blogs from the 2004 US presidential election. In Adamic & Glance (2005) it is found that there is an underlying structure with $K = 2$ clusters corresponding to the Republican and Democratic parties. Hence we choose $K = 2$ and $\beta = 1$.
• *Migration* (Perry, 2003) reports the number of people that migrated between pairs of counties in the US during 1995-2000. It involves $n = 3,075$ countries and $|\mathcal{E}| = 721,432$ directed edges after obtaining the largest weakly connected component. We choose $K = 10$ and $\beta = 9$, following (Cucuringu et al., 2020a). Since the original digraph has entries extremely large, to cope with these outliers, we preprocess the input network by

$$\mathbf{A}_{i,j} = \frac{\mathbf{A}_{i,j}}{\mathbf{A}_{i,j} + \mathbf{A}_{j,i}} \mathbb{1}(\mathbf{A}_{i,j} > 0), \forall i, j \in \{1, \cdots, n\}, \tag{13}$$

which follows the preprocessing of (Cucuringu et al., 2020a). The results for not doing this preprocessing is provided in Table 11.
• *WikiTalk* (Leskovec et al., 2010) contains all users and discussion from the inception of Wikipedia until Jan. 2008. The $n = 2,388,953$ nodes in the network represent Wikipedia users and a directed edge from node $v_i$ to node $v_j$ denotes that user $i$ edited at least once a talk page of user $j$. There are $|\mathcal{E}| = 5,018,445$ edges. We choose $K = 10$ clusters among candidates $\{2, 3, 5, 6, 8, 10\}$, and $\beta = 10$.
• *Lead-Lag* (Bennett et al., 2021) contains yearly lead-lag matrices from 269 stocks from 2001 to 2019. We choose $K = 10$ clusters based on the GICS industry sectors (Phillips & Ormsby, 2016), and choose $\beta = 3$ to emphasize the top three pairs of imbalance values. The lead-lag matrices are built from time series of daily price log returns, as detailed in Bennett et al. (2021). The lead-lag metric for entry $(i, j)$ in the network encodes a measure of the extent to which stock $i$ leads stock $j$, and is obtained by applying a functional that computes the signed normalized area under the curve (auc) of the standard cross-correlation function (ccf). The resulting matrix is skew-symmetric, and entry $(i, j)$ quantifies the extent to which stock $i$ leads or lags stocks $j$, thus leading to a directed network interpretation. Starting from the skew-symmetric matrix, we further convert negative entries to zero, so that the resulting digraph can be directly fed into other methods; note that this step does not throw away any information, and is pursued only to render the representation of the digraph consistent with the format expected by all methods compared, including DIGRAC. Note that the statistics given in Table 3 are averaged over the 19 years.

Table 3: Summary statistics for the real-world networks.

| data set | $n$ | $|\mathcal{E}|$ | density | weighted | $|\mathcal{E}^r|$ | $\frac{|\mathcal{E}^r|}{|\mathcal{E}|}(\%)$ |
|---|---|---|---|---|---|---|
| *Telegram* | 245 | 8,912 | $1.28 \cdot 10^{-2}$ | True | 1,572 | 17.64 |
| *Blog* | 1,222 | 19,024 | $1.49 \cdot 10^{-1}$ | True | 4,617 | 24.27 |
| *Migration* | 3,075 | 721,432 | $7.63 \cdot 10^{-2}$ | True | 351,100 | 48.67 |
| *WikiTalk* | 2,388,953 | 5,018,445 | $8.79 \cdot 10^{-7}$ | False | 723,526 | 14.42 |
| *Lead-Lag* | 269 | 29,159 | $4.04 \cdot 10^{-1}$ | True | 0.00 | 0.00 |

As input features, after obtaining eigenvectors from Hermitian matrices constructed as in Cucuringu et al. (2020a), we standardize each column vector so that it has mean zero and variance one. We use these features for all GNN methods except MagNet, since MagNet has its own way of generating random features of dimension one.

### B.4 HYPERPARAMETERS

We conduct hyperparmeter selection via a greedy search. To explain the details, consider for example the following synthetic data setting: DSBM with 1000 nodes, 5 clusters, $\rho = 1$, and $p = 0.02$, without ambient nodes under different hyperparameter settings. By default, we use the loss function $\mathcal{L}^{\text{sort}}_{\text{vol\_sum}}$, $d = 32$ hidden units, hop $h = 2$, and no seed nodes. Instead of a grid search, we tune hyperparameters according to what performs the best in the default setting of the respective GNN method. The procedure starts with a random setting. For the next iteration, the hyperparameters are set to the current best setting (based on the last iteration), independently. For example, if we start with $a = 1, b = 2, c = 3$, and we find that under this default setting, the best $a$ (when fixing $b = 2, c = 3$) is 2 and the best $b$ (when fixing $a = 1, c = 3$) is 3, and the best $c$ is 3 (when fixing $a = 1, b = 2$), then for the next iteration, we set $a = 2, b = 3, c = 3$. If two settings give similar results, we choose the simpler setting, for example, the smaller hop size. When we reach a local optimum, we stop searching. Indeed, just a few iterations (less than five) were required for us to find the current setting, as DIGRAC tends to be robust to most hyperparameters.

Figure 8, 9 and 10 are plots corresponding to the same setting but for three different meta-graph structures, namely the complete meta-graph structure, the cycle structure but with ambient nodes, and the complete structure with ambient nodes, respectively.

In theory, more hidden units give better expressive power. To reduce complexity, we use 32 hidden units throughout, which seems to have desirable performance. We observe that for low-noise regimes, more hidden units actually hurt performance. We can draw a similar conclusion about the hyperparameter selection. In terms of $\tau$, DIGRAC seems to be robust to different choices. Therefore, we use $\tau = 0.5$ throughout.

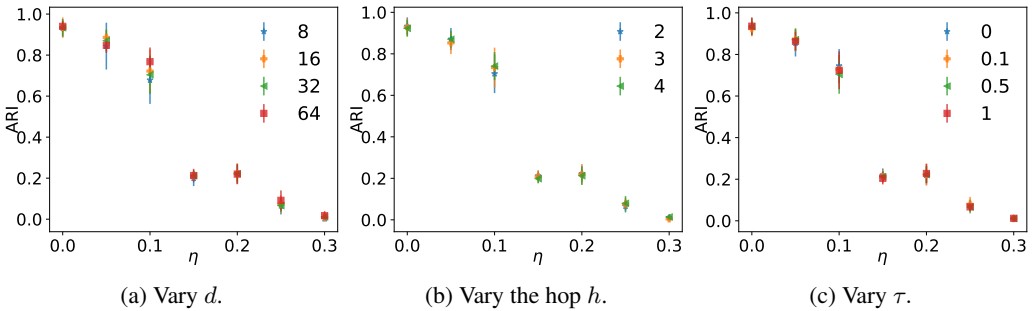

(a) Vary $d$.    (b) Vary the hop $h$.    (c) Vary $\tau$.

Figure 8: Hyperparameter analysis on different hyperparameter settings on the **complete** DSBM with 1000 nodes, 5 clusters, $\rho = 1$, and $p = 0.02$ **without** ambient nodes.

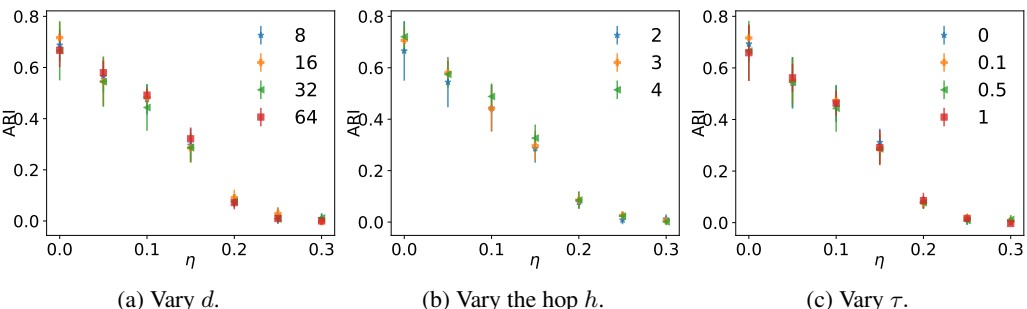

(a) Vary $d$.    (b) Vary the hop $h$.    (c) Vary $\tau$.

Figure 9: Hyperparameter analysis on different hyperparameter settings on the **complete** DSBM with 1000 nodes, 5 clusters, $\rho = 1$, and $p = 0.02$ **with** ambient nodes.

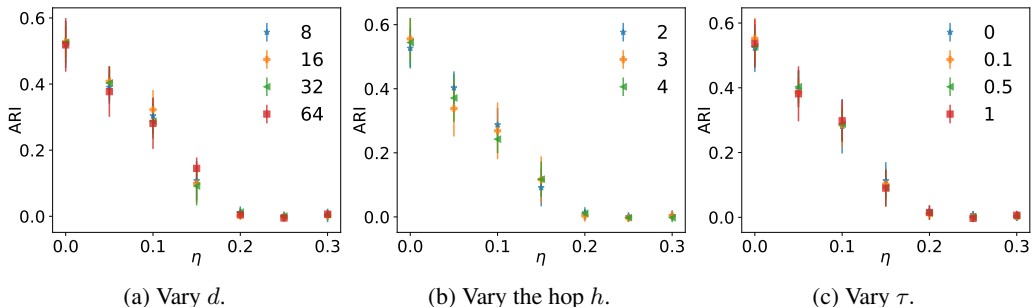

(a) Vary $d$.  (b) Vary the hop $h$.  (c) Vary $\tau$.

Figure 10: Hyperparameter analysis on different hyperparameter settings on the **cycle** DSBM with 1000 nodes, 5 clusters, $\rho = 1$, and $p = 0.02$ **with** ambient nodes.

### B.5 USE OF SEED NODES IN A SEMI-SUPERVISED MANNER

#### B.5.1 SUPERVISED LOSS

For seed nodes in $\mathcal{V}^{\text{seed}}$, similar to the loss function in Tian et al. (2019), we use as a supervised loss function the sum of a cross-entropy loss and a triplet loss. The cross-entropy loss is given by

$$\mathcal{L}_{\text{CE}} = -\frac{1}{|\mathcal{V}^{\text{seed}}|} \sum_{v_i \in \mathcal{V}^{\text{seed}}} \sum_{k=1}^{K} \mathbb{1}(v_i \in \mathcal{C}_k) \log\left((\mathbf{p}_i)_k\right), \tag{14}$$

where $\mathbb{1}$ is the indicator function, $\mathcal{C}_k$ denotes the $k^{th}$ cluster, and $(\mathbf{p}_i)_k$ denotes the $k^{th}$ entry of probability vector $(\mathbf{p}_i)$. With the function $L : \mathbb{R}^2 \to \mathbb{R}$ given by $L(x, y) = [x - y]_+$ (where the subscript $+$ indicates taking the maximum of the expression value and 0), the triplet loss is defined as

$$\mathcal{L}_{\text{triplet}} = \frac{1}{|\mathcal{S}|} \sum_{(v_i, v_j, v_k) \in \mathcal{S}} L(\text{CS}(\mathbf{z}_i, \mathbf{z}_j), \text{CS}(\mathbf{z}_i, \mathbf{z}_k)), \tag{15}$$

where $\mathcal{S} \subseteq \mathcal{V}^{\text{seed}} \times \mathcal{V}^{\text{seed}} \times \mathcal{V}^{\text{seed}}$ is a set of node triplets: $v_i$ is an anchor seed node, and $v_j$ is a seed node from the same cluster as the anchor, while $v_k$ is from a different cluster; and $\text{CS}(\mathbf{z}_i, \mathbf{z}_j)$ is the cosine similarity of the embeddings of nodes $v_i$ and $v_j$. We choose cosine similarity so as to avoid sensitivity to the magnitude of the embeddings. The triplet loss is designed so that, given two seed nodes from the same cluster and one seed node from a different cluster, the respective embeddings of the pairs from different clusters should be farther away than the embedding of the pair within the same cluster.

We then consider the weighted sum $\mathcal{L}_{\text{CE}} + \gamma_t \mathcal{L}_{\text{triplet}}$ as the supervised part of the loss function for DIGRAC, for some parameter $\gamma_t > 0$. The parameter $\gamma_t$ arises as follows. The cosine similarity between two randomly picked vectors in $d$ dimensions is bounded by $\sqrt{\ln(d)/d}$ with high probability. In our experiments $d = 32$, and $\sqrt{\ln(2d)/(2d)} \approx 0.25$. In contrast, for fairly uniform clustering, the cross-entropy loss grows like $\log n$, which in our experiments ranges between 3 and 17. Thus some balancing of the contribution is required. Following (Tian et al., 2019), we choose $\gamma_t = 0.1$ in our experiments.

#### B.5.2 OVERALL OBJECTIVE FUNCTION

By combining Eq. (14), Eq. (15), and Eq. (3), our objective function for semi-supervised training with known seed nodes minimizes

$$\mathcal{L} = \mathcal{L}_{\text{vol\_sum}}^{\text{sort}} + \gamma_s(\mathcal{L}_{\text{CE}} + \gamma_t \mathcal{L}_{\text{triplet}}), \tag{16}$$

where $\gamma_s, \gamma_t > 0$ are weights for the supervised part of the loss and triplet loss within the supervised part, respectively. We set $\gamma_s = 50$ as we want our model to perform well on seed nodes. The weights could be tuned depending on how important each term is perceived to be.

### B.6 TRAINING

For all synthetic data, we train DIGRAC with a maximum of 1000 epochs, and stop training when no gain in validation performance is achieved for 200 epochs (early-stopping). For real-world data, no "ground-truth" labels are available; we use all nodes to train and stop training when the training loss does not decrease for 200 epochs, or when we reach the maximum number of epochs, 1000.

For the two-layer MLP, we do not have a bias term for each layer, and we use Rectified Linear Unit (ReLU) followed by a dropout layer with 0.5 dropout probability between the two layers, following (Tian et al., 2019). We use Adam (Kingma & Ba, 2014) as the optimizer and $\ell_2$ regularization with weight decay $5 \cdot 10^{-4}$ to avoid overfitting. We use as learning rate 0.01 throughout.

### B.7 IMPLEMENTATION DETAILS FOR THE COMPARISON METHODS

In our experiments, we compare DIGRAC against five spectral methods and five GNN-based supervised methods on synthetic data, and spectral methods on real data. The reason we are not able to compare DIGRAC with the above GNNs on these data sets is due to the fact that these data sets do not have labels, which are required by the other GNN methods. We use the same hyperparameter settings stated in these papers. Data splits for all models are the same; the comparison GNNs are trained with 80% nodes under label supervision.

For MagNet, we use $q = 0.25$ for the phase matrix as in Zhang et al. (2021b), because it is mentioned that $q = 0.25$ lays the most emphasis on directionality, which is our main focus in this paper. Code for MagNet is from `https://github.com/matthew-hirn/magnet`. We use the code from `https://github.com/flyingtango/DiGCN/blob/main/code/digcn.py` to obtain the log of probability matrix $\mathbf{P}$ for the methods DiGCN and DiGCN_app. The only difference between these two methods is whether or not to use approximate Laplacian based on personalized PageRank. The "adj_type" options for them correspond to "or" and "appr", respectively.

For DiGCN_ib, we use the code from `https://github.com/flyingtango/DiGCN/blob/main/code/digcn_ib.py` with option "adj_type" equals "ib". As a recommended option in (Tong et al., 2020a), we use three layers for DiGCN_ib and two layers for DiGCN and DiGCN_app. All other settings are the same as in the original paper (Tong et al., 2020a).

### B.8 ALGORITHM AND COMPLEXITY ANALYSIS

To avoid computationally expensive and space unfriendly matrix operations, as described in Eq. 4 in the main text, DIGRAC uses an efficient sparsity-aware implementation, described in Algorithm 1, without explicitly calculating the sets of powers $\mathcal{A}^{s,h}$ and $\mathcal{A}^{t,h}$. We omit the subscript $\mathcal{V}$ for ease of notation. The algorithm is efficient in the sense that it takes sparse matrices as input, and never explicitly computes a multiplication of two $n \times n$ matrices. Therefore, for input feature dimension $d_{\text{in}}$ and hidden dimension $d$, if $d' = \max(d_{\text{in}}, d) \ll n$, time and space complexity of DIMPA, and implicitly DIGRAC, is $\mathcal{O}(|\mathcal{E}|d'h^2 + 2nd'K)$ and $\mathcal{O}(2|\mathcal{E}| + 4nd' + nK)$, respectively (Harrison & Joseph, 2018; Greiner & Jacob, 2010).

Indeed, it is a current shortcoming of DIGRAC that it does not scale well to very large networks; however, this limitation is also shared by all the GNN competitors compared against in the paper, and some of the spectral methods. DIGRAC scales well in the sense that when the underlying network is sparse, the sparsity is preserved throughout the pipeline. In contrast, Bi_sym and DD_sym (Satuluri & Parthasarathy, 2011) construct derived dense matrices for manipulation, rendering the methods no longer scalable. These methods resulted in N/A values in Table 1 in the main text. For large-scale networks, DIMPA is amenable to a minibatch version using neighborhood sampling, similar to the minibatch forward propagation algorithm in Hamilton et al. (2017); Markowitz et al. (2021). We are also aware of a framework (Fey et al., 2021) for scaling up graph neural networks automatically, where theoretical guarantees are provided, and ideas there will be exploited in future. We expect that the theoretical guarantees could be adapted to our situation.

---

**Algorithm 1:** Weighted Multi-Hop Neighbor Aggregation (DIMPA).

---

**Input**   : (Sparse) row-normalized adjacency matrices $\overline{\mathbf{A}}^s, \overline{\mathbf{A}}^t$; initial hidden representations
$\mathbf{H}^s, \mathbf{H}^t$; hop $h(h \geq 2)$; lists of scalar weights
$\Omega^s = (\omega_{\mathbf{M}}^s, \mathbf{M} \in \mathcal{A}^{s,h}), \Omega^t = (\omega_{\mathbf{M}}^t, \mathbf{M} \in \mathcal{A}^{t,h})$.
**Output** : Vector representations $\mathbf{z}_i$ for all $v_i \in \mathcal{V}$ given by $\mathbf{Z}$.

$\qquad \tilde{\mathbf{X}}^s \leftarrow \overline{\mathbf{A}}^s \mathbf{H}^s; \qquad\qquad\qquad\qquad\qquad \tilde{\mathbf{X}}^t \leftarrow \overline{\mathbf{A}}^t \mathbf{H}^t;$
$\qquad \mathbf{Z}^s \leftarrow \Omega^s[0] \cdot \mathbf{H}^s + \Omega^s[1] \cdot \tilde{\mathbf{X}}^s; \qquad\qquad \mathbf{Z}^t \leftarrow \Omega^t[0] \cdot \mathbf{H}^t + \Omega^t[1] \cdot \tilde{\mathbf{X}}^t;$
**for** $i \leftarrow 2$ **to** $h$ **do**
$\qquad \tilde{\mathbf{X}}^s \leftarrow \overline{\mathbf{A}}^s \tilde{\mathbf{X}}^s; \qquad\qquad\qquad\qquad\qquad \tilde{\mathbf{X}}^t \leftarrow \overline{\mathbf{A}}^t \tilde{\mathbf{X}}^t;$
$\qquad \mathbf{Z}^s \leftarrow \mathbf{Z}^s + \Omega^s[i] \cdot \tilde{\mathbf{X}}^s; \qquad\qquad\qquad \mathbf{Z}^t \leftarrow \mathbf{Z}^t + \Omega^t[i] \cdot \tilde{\mathbf{X}}^t;$
**end**
$\mathbf{Z} = \mathrm{CONCAT}\,(\mathbf{Z}^s, \mathbf{Z}^t);$

---

## C  MORE RESULTS ON SYNTHETIC DATA

### C.1  AN ADDITIONAL META-GRAPH STRUCTURE

Recall that the Directed Stochastic Block Models used in our experiments depend on a meta-graph adjacency matrix $\mathbf{F}$ and a filled version of it, $\tilde{\mathbf{F}}$, for some number of clusters, $K$, and noise level $\eta \leq 0.5$. The meta-graph adjacency matrix $\mathbf{F}$ is generated from some meta-graph structure, called $\mathcal{M}$. Based on $\mathcal{M}$, the filled meta-graph $\tilde{\mathbf{F}}$ replaces every zero in $\mathbf{F}$ that is not part of the imbalance structure with 0.5, independently of the choice of $\eta$. It is the filled meta-graph $\tilde{\mathbf{F}}$ which we feed into the DSBM generation process. The filled meta-graph creates a number of *ambient nodes* which correspond to entries which are not part of the imbalance structure and thus are not part of a meaningful cluster; the set of *ambient nodes* is also called the *ambient cluster*.

Here, we introduce an additional meta-graph structure, called "multipartite", following (Elliott et al., 2019). First, when there are no ambient nodes: we divide the index set into three sets; setting $i_1 = \lfloor \frac{K}{9} \rfloor, i_2 = \lfloor \frac{3K}{9} \rfloor + i_1$, let

$$
\begin{aligned}
\mathbf{F}_{k,l} \quad = \quad & (1-\eta)\mathbb{1}(k < i_1, i_1 \leq l < i_2) + \eta\mathbb{1}(i_1 \leq k < i_2, l \geq i_2) \\
& + (1-\eta)\mathbb{1}(k \geq i_2, i_1 \leq l < i_2) + \eta\mathbb{1}(i_1 \leq k < i_2, l < i_1).
\end{aligned}
$$

When we have ambient nodes, the construction involves two steps, with the first step the same as the above but with the following changes: divide the indices into three sets, with set boundaries given by $i_1 = \lfloor \frac{K-1}{9} \rfloor, i_2 = \lfloor \frac{3(K-1)}{9} \rfloor + i_1$. The second step is to assign 0 (respectively, 0.5) to the last row and the last column of $\mathbf{F}$ (respectively, $\tilde{\mathbf{F}}$).

### C.2  ADDITIONAL COMPARISON PLOTS AND ANALYSIS

Figure 11 compares the numerical performance of DIGRAC with other methods except InfoMap, which performs rather poorly in our synthetic experiments in the main text, on four more settings of synthetic data, namely, a cycle structure with three clusters, a complete structure with ten clusters, a multipartite structure with ten clusters, and a star structure with five clusters. Indeed, Infomap clusters all nodes into one big cluster for all of our synthetic experiments. Considering the results in Section 5 and Figure 11, we remark that DIGRAC gives state-of-the-art results on a wide range of network densities and noise levels, on different scales of the networks, and with different meta-graph structures, whether or not ambient nodes exist.

Note that the multipartite, the cycle and the star settings correspond to the intuition behind (Tong et al., 2020a) which assumes that nodes are similar if their set of $k^{th}$-order neighborhoods are similar; here the second-order neighborhoods are similar by design. For networks with underlying meta-graph structure "star", "cycle" or "multipartite", clusters could be determined by grouping nodes that share similar in-neighbors and out-neighbors together, which aligns well with the second-order proximity used in DGCN and DiGCN_ib from (Tong et al., 2020b). Therefore, these methods are naturally well-suited for dealing with the such synthetic data. We also note that although

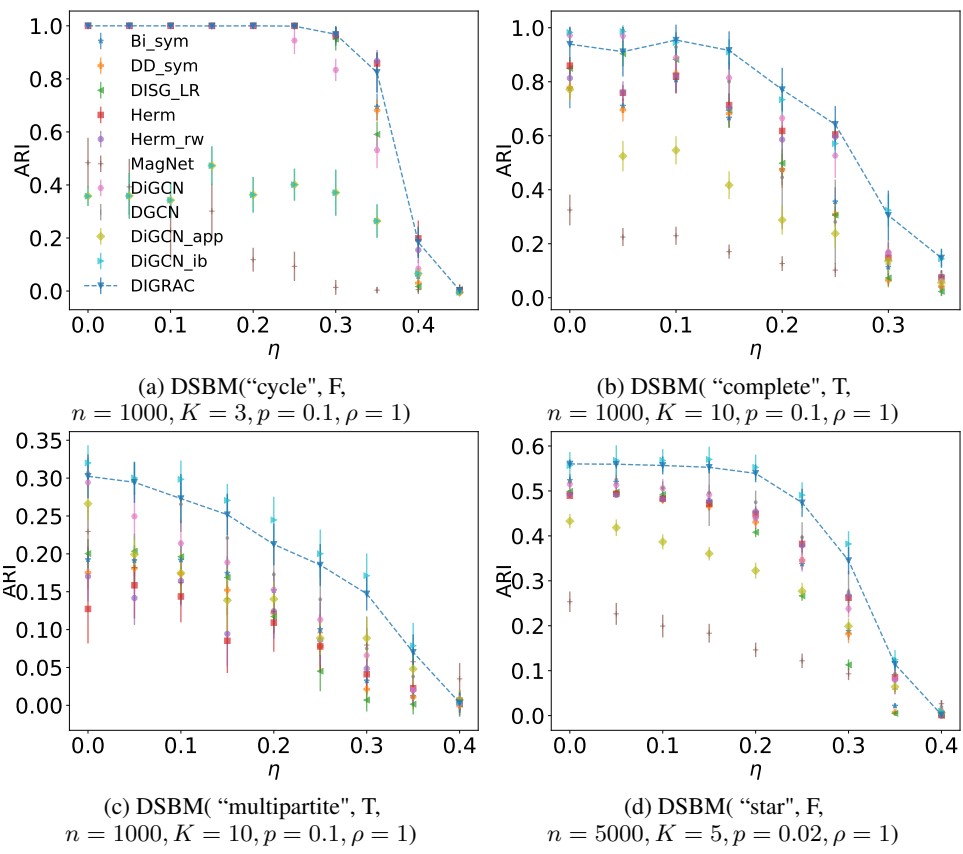

Figure 11: Node clustering test ARI comparison on four additional synthetic data sets. Dashed lines highlight DIGRAC's performance. Error bars are given by one standard error. Abbreviations for all the methods are given in Section 4 in the main text.

DIGRAC does not explicitly use second-order proximity, it can achieve comparable performance with DGCN and DiGCN_ib. This indicates DIGRAC's flexibility to adapt to directed networks with different underlying topologies, without explicitly utilizing higher-order proximity. On the other hand, DiGCN_ib is fully supervised, and takes much more space and time to implement, than DIGRAC. This is partially due to the use of the so-called *inception blocks* in DiGCN_ib, where multi-scale directed structure features are encoded and fused with a fusion function. As stated in Tong et al. (2020a), the worst space complexity is $\mathcal{O}(k'n^2)$, where $k'$ is the order of proximity to consider (we use $k' = 2$ throughout). The eigenvalue decomposition in the preprocessing step is $\mathcal{O}(n^3)$. We also remark that the approximate Laplacian based on personalized PageRank, when no inception blocks are used, performs no better than the simpler implementation without the approximation. We conclude that overall DIGRAC is a fast method for general directed clustering when directionality is the main signal, which performs as well as custom-tailored methods when the proximity neighborhood heuristic holds, while outperforming all tested methods on the complete meta-graph, where the proximity neighborhood heuristic does not hold.

## D  ADDITIONAL RESULTS ON REAL-WORLD DATA

### D.1  EXTENDED RESULT TABLES

Tables 4, 5, 6 and 7 provide a detailed comparison of DIGRAC with spectral methods. Since no labeling information is available and all of the other competing GNN methods require labels, we do not compare DIGRAC with them on these real data sets.

In Tables 4, 5, 6 and 7, we report 12 combinations of global imbalance scores by data set. The naming convention of these imbalance scores is provided in Table 2. To assess how balanced our recovered clusters are in terms of sizes, we also report the size ratio, which is defined as the size of the largest predicted cluster to the smallest one, and the standard deviation of sizes, size std, in order to show how varied the sizes of predicted clusters are. For a relatively balanced clustering, we expect the latter two terms to be small.

Table 4: Performance comparison on *Telegram*. The best is marked in **bold red** and the second best is marked in underline blue.

| Metric/Method | InfoMap | Bi_sym | DD_sym | DISG_LR | Herm | Herm_rw | DIGRAC |
|---|---|---|---|---|---|---|---|
| $\mathcal{O}^{sort}_{vol\_sum}$ | 0.04±0.00 | 0.21±0.00 | 0.21±0.00 | 0.21±0.01 | 0.20±0.01 | 0.14±0.00 | **0.32±0.01** |
| $\mathcal{O}^{sort}_{vol\_min}$ | 0.47±0.00 | 0.67±0.00 | 0.61±0.00 | 0.66±0.02 | 0.66±0.02 | 0.19±0.00 | **0.79±0.06** |
| $\mathcal{O}^{sort}_{vol\_max}$ | 0.03±0.00 | 0.20±0.00 | 0.20±0.00 | 0.20±0.01 | 0.19±0.01 | 0.12±0.00 | **0.29±0.01** |
| $\mathcal{O}^{sort}_{plain}$ | **1.00±0.00** | 0.80±0.00 | 0.75±0.00 | 0.78±0.03 | 0.76±0.04 | 0.59±0.00 | 0.96±0.01 |
| $\mathcal{O}^{std}_{vol\_sum}$ | 0.01±0.00 | 0.26±0.00 | 0.26±0.00 | 0.26±0.01 | 0.25±0.02 | **0.35±0.00** | 0.28±0.01 |
| $\mathcal{O}^{std}_{vol\_min}$ | 0.16±0.00 | **0.84±0.00** | 0.76±0.00 | 0.82±0.03 | 0.82±0.03 | 0.49±0.00 | 0.73±0.03 |
| $\mathcal{O}^{std}_{vol\_max}$ | 0.01±0.00 | 0.25±0.00 | 0.25±0.00 | 0.25±0.01 | 0.24±0.02 | **0.29±0.00** | 0.25±0.01 |
| $\mathcal{O}^{std}_{plain}$ | 0.68±0.00 | **1.00±0.00** | 0.94±0.00 | 0.98±0.04 | 0.95±0.04 | 0.99±0.00 | 0.90±0.05 |
| $\mathcal{O}^{naive}_{vol\_sum}$ | 0.01±0.00 | 0.26±0.00 | 0.26±0.00 | 0.26±0.01 | 0.25±0.02 | 0.23±0.00 | **0.27±0.01** |
| $\mathcal{O}^{naive}_{vol\_min}$ | 0.11±0.00 | **0.84±0.00** | 0.76±0.00 | 0.82±0.03 | 0.82±0.03 | 0.32±0.00 | 0.72±0.04 |
| $\mathcal{O}^{naive}_{vol\_max}$ | 0.00±0.00 | **0.25±0.00** | **0.25±0.00** | **0.25±0.01** | 0.24±0.02 | 0.20±0.00 | 0.24±0.01 |
| $\mathcal{O}^{naive}_{plain}$ | 0.63±0.00 | **1.00±0.00** | 0.94±0.00 | 0.98±0.04 | 0.95±0.04 | 0.99±0.00 | 0.89±0.06 |
| size ratio | 24.750 | 242.000 | 242.000 | 242.000 | 242.00 | 53 | **3.090** |
| size std | 35.57 | 104.360 | 104.360 | 104.360 | 104.360 | 63.460 | **26.39** |

Table 5: Performance comparison on *Blog*. The best is marked in **bold red** and the second best is marked in underline blue.

| Metric/Method | InfoMap | Bi_sym | DD_sym | DISG_LR | Herm | Herm_rw | DIGRAC |
|---|---|---|---|---|---|---|---|
| $\mathcal{O}^{sort}_{vol\_sum}$ | 0.07±0.00 | 0.07±0.00 | 0.00±0.00 | 0.05±0.00 | 0.37±0.00 | 0.00±0.00 | **0.44±0.00** |
| $\mathcal{O}^{sort}_{vol\_min}$ | 0.02±0.00 | 0.33±0.00 | 0.05±0.00 | 0.31±0.00 | 0.78±0.01 | **0.89±0.00** | 0.76±0.00 |
| $\mathcal{O}^{sort}_{vol\_max}$ | 0.05±0.00 | 0.05±0.00 | 0.00±0.00 | 0.04±0.00 | 0.26±0.00 | 0.00±0.00 | **0.40±0.00** |
| $\mathcal{O}^{sort}_{plain}$ | **1.00±0.00** | 0.33±0.00 | 0.05±0.00 | 0.31±0.00 | 0.78±0.01 | 0.89±0.00 | 0.76±0.00 |
| $\mathcal{O}^{std}_{vol\_sum}$ | 0.00±0.00 | 0.07±0.00 | 0.00±0.00 | 0.05±0.00 | 0.37±0.00 | 0.00±0.00 | **0.44±0.00** |
| $\mathcal{O}^{std}_{vol\_min}$ | 0.00±0.00 | 0.33±0.00 | 0.05±0.00 | 0.31±0.00 | 0.78±0.01 | **0.89±0.00** | 0.76±0.00 |
| $\mathcal{O}^{std}_{vol\_max}$ | 0.00±0.00 | 0.05±0.00 | 0.00±0.00 | 0.04±0.00 | 0.26±0.00 | 0.00±0.00 | **0.40±0.00** |
| $\mathcal{O}^{std}_{plain}$ | 0.73±0.00 | 0.33±0.00 | 0.05±0.00 | 0.31±0.00 | 0.78±0.01 | **0.89±0.00** | 0.76±0.00 |
| $\mathcal{O}^{naive}_{vol\_sum}$ | 0.00±0.00 | 0.07±0.00 | 0.00±0.00 | 0.05±0.00 | 0.37±0.00 | 0.00±0.00 | **0.44±0.00** |
| $\mathcal{O}^{naive}_{vol\_min}$ | 0.00±0.00 | 0.33±0.00 | 0.05±0.00 | 0.31±0.00 | 0.78±0.01 | **0.89±0.00** | 0.76±0.00 |
| $\mathcal{O}^{naive}_{vol\_max}$ | 0.00±0.00 | 0.05±0.00 | 0.00±0.00 | 0.04±0.00 | 0.26±0.00 | 0.00±0.00 | **0.40±0.00** |
| $\mathcal{O}^{naive}_{plain}$ | 0.76±0.00 | 0.33±0.00 | 0.05±0.00 | 0.31±0.00 | 0.78±0.01 | **0.89±0.00** | 0.76±0.00 |
| size ratio | **1.270** | 8.700 | 2.450 | 6.100 | 11.93 | 44.26 | 1.860 |
| size std | **64.50** | 485 | 256.200 | 439 | 516.500 | 584 | 183.20 |

Tables 4, 5, 6, 7, 8, 9 and 10 reveal that DIGRAC provides competitive global imbalance scores in all of the 12 objectives introduced, and across all the real data sets, usually outperforming all the other methods. Among the tables, Table 10 provides results in terms of the distance to the best yearly performance, averaged across the 19 years; DIGRAC usually outperforms all the other methods across all the years. Note that Bi_sym and DD_sym are not able to generate results for *WikiTalk*, as large $n \times n$ matrix multiplication with its transpose causes memory issue, when $n = 2,388,953$. Small values of the size ratio and size standard deviation suggest that the normalization in the loss function penalizes tiny clusters, and that DIGRAC tends to predict balanced cluster sizes.

Table 6: Performance comparison on *Migration*. The best is marked in **bold red** and the second best is marked in underline blue. InfoMap results are omitted here as it predicts a single huge cluster and could not generate imbalance results.

| Metric/Method | Bi_sym | DD_sym | DISG_LR | Herm | Herm_rw | DIGRAC |
|---|---|---|---|---|---|---|
| $\mathcal{O}^{sort}_{vol\_sum}$ | 0.03±0.00 | 0.01±0.00 | 0.02±0.00 | 0.04±0.00 | 0.02±0.00 | **0.05±0.00** |
| $\mathcal{O}^{sort}_{vol\_min}$ | **0.19±0.00** | 0.08±0.00 | 0.08±0.00 | 0.15±0.02 | 0.05±0.00 | 0.18±0.03 |
| $\mathcal{O}^{sort}_{vol\_max}$ | 0.03±0.00 | 0.01±0.00 | 0.01±0.00 | 0.03±0.00 | 0.02±0.00 | **0.04±0.00** |
| $\mathcal{O}^{sort}_{plain}$ | 0.24±0.00 | 0.20±0.00 | 0.17±0.00 | 0.40±0.01 | **0.49±0.06** | 0.29±0.04 |
| $\mathcal{O}^{std}_{vol\_sum}$ | 0.01±0.00 | 0.01±0.00 | 0.01±0.00 | 0.02±0.00 | 0.02±0.00 | **0.04±0.01** |
| $\mathcal{O}^{std}_{vol\_min}$ | 0.10±0.00 | 0.05±0.00 | 0.05±0.00 | 0.08±0.01 | 0.04±0.00 | **0.16±0.03** |
| $\mathcal{O}^{std}_{vol\_max}$ | 0.01±0.00 | 0.01±0.00 | 0.01±0.00 | 0.01±0.00 | 0.01±0.00 | **0.03±0.01** |
| $\mathcal{O}^{std}_{plain}$ | 0.13±0.00 | 0.12±0.00 | 0.11±0.00 | 0.20±0.01 | 0.20±0.01 | **0.26±0.01** |
| $\mathcal{O}^{naive}_{vol\_sum}$ | 0.01±0.00 | 0.01±0.00 | 0.01±0.00 | 0.02±0.00 | 0.01±0.00 | **0.04±0.01** |
| $\mathcal{O}^{naive}_{vol\_min}$ | 0.09±0.00 | 0.04±0.00 | 0.04±0.00 | 0.08±0.01 | 0.01±0.00 | **0.16±0.03** |
| $\mathcal{O}^{naive}_{vol\_max}$ | 0.01±0.00 | 0.00±0.00 | 0.01±0.00 | 0.01±0.00 | 0.00±0.00 | **0.03±0.01** |
| $\mathcal{O}^{naive}_{plain}$ | 0.12±0.00 | 0.10±0.00 | 0.08±0.00 | 0.19±0.00 | 0.19±0.03 | **0.26±0.01** |
| size ratio | 7.780 | 6.070 | **4.360** | 36.05 | 1035.90 | 4.420 |
| size std | 135.210 | 132.76 | **103.43** | 335.790 | 353.060 | 264.500 |

Table 7: Performance comparison on *WikiTalk*. The best is marked in **bold red** and the second best is marked in underline blue. InfoMap results are omitted here as its large number of predicted clusters leads to memory error in imbalance calculation.

| Metric/Method | DISG_LR | Herm | Herm_rw | DIGRAC |
|---|---|---|---|---|
| $\mathcal{O}^{sort}_{vol\_sum}$ | 0.18±0.03 | 0.15±0.02 | 0.00±0.00 | **0.24±0.05** |
| $\mathcal{O}^{sort}_{vol\_min}$ | 0.10±0.03 | 0.22±0.05 | 0.26±0.00 | **0.28±0.13** |
| $\mathcal{O}^{sort}_{vol\_max}$ | 0.16±0.03 | 0.09±0.01 | 0.00±0.00 | **0.19±0.04** |
| $\mathcal{O}^{sort}_{plain}$ | 0.87±0.08 | 0.99±0.01 | 0.98±0.00 | **1.00±0.00** |
| $\mathcal{O}^{std}_{vol\_sum}$ | **0.17±0.04** | 0.06±0.01 | 0.01±0.00 | 0.14±0.02 |
| $\mathcal{O}^{std}_{vol\_min}$ | 0.09±0.02 | 0.09±0.02 | **0.27±0.00** | 0.18±0.08 |
| $\mathcal{O}^{std}_{vol\_max}$ | **0.15±0.04** | 0.04±0.00 | 0.00±0.00 | 0.11±0.02 |
| $\mathcal{O}^{std}_{plain}$ | 0.72±0.03 | 0.70±0.05 | **0.98±0.00** | 0.84±0.06 |
| $\mathcal{O}^{naive}_{vol\_sum}$ | 0.10±0.02 | 0.04±0.00 | 0.00±0.00 | **0.12±0.01** |
| $\mathcal{O}^{naive}_{vol\_min}$ | 0.06±0.03 | 0.07±0.02 | **0.26±0.00** | 0.15±0.07 |
| $\mathcal{O}^{naive}_{vol\_max}$ | **0.09±0.02** | 0.03±0.00 | 0.00±0.00 | **0.09±0.01** |
| $\mathcal{O}^{naive}_{plain}$ | 0.64±0.04 | 0.61±0.04 | **0.98±0.00** | 0.76±0.06 |
| size ratio | 1190162.25 | 2217434.50 | **250.48** | 71765.14 |
| size std | 713813.72 | 660060.33 | 657941.88 | **643220.37** |

Table 8: Performance comparison on *Lead-Lag* for year 2015. The best is marked in **bold red** and the second best is marked in underline blue. InfoMap results are omitted here as it usually predicts a single huge cluster and could not generate imbalance results.

| Metric/Method | Bi_sym | DD_sym | DISG_LR | Herm | Herm_rw | DIGRAC |
|---|---|---|---|---|---|---|
| $\mathcal{O}^{sort}_{vol\_sum}$ | 0.07±0.00 | 0.07±0.00 | 0.06±0.00 | 0.07±0.00 | 0.06±0.01 | **0.15±0.00** |
| $\mathcal{O}^{sort}_{vol\_min}$ | **0.53±0.06** | 0.50±0.02 | 0.45±0.07 | 0.50±0.03 | 0.46±0.06 | 0.50±0.02 |
| $\mathcal{O}^{sort}_{vol\_max}$ | 0.07±0.00 | 0.06±0.00 | 0.06±0.00 | 0.06±0.00 | 0.06±0.00 | **0.15±0.01** |
| $\mathcal{O}^{sort}_{plain}$ | 0.65±0.03 | **0.67±0.03** | 0.59±0.03 | 0.65±0.03 | 0.65±0.02 | 0.55±0.07 |
| $\mathcal{O}^{std}_{vol\_sum}$ | 0.04±0.00 | 0.04±0.00 | 0.04±0.00 | 0.04±0.00 | 0.04±0.00 | **0.11±0.02** |
| $\mathcal{O}^{std}_{vol\_min}$ | 0.27±0.03 | 0.27±0.02 | 0.24±0.02 | 0.27±0.02 | 0.26±0.04 | **0.35±0.04** |
| $\mathcal{O}^{std}_{vol\_max}$ | 0.03±0.00 | 0.03±0.00 | 0.03±0.00 | 0.03±0.00 | 0.03±0.00 | **0.10±0.02** |
| $\mathcal{O}^{std}_{plain}$ | 0.39±0.02 | 0.39±0.01 | 0.37±0.02 | 0.39±0.02 | **0.40±0.02** | 0.38±0.04 |
| $\mathcal{O}^{naive}_{vol\_sum}$ | 0.03±0.00 | 0.03±0.00 | 0.03±0.00 | 0.03±0.00 | 0.03±0.00 | **0.08±0.03** |
| $\mathcal{O}^{naive}_{vol\_min}$ | 0.20±0.02 | 0.20±0.02 | 0.17±0.03 | 0.20±0.02 | 0.20±0.03 | **0.25±0.08** |
| $\mathcal{O}^{naive}_{vol\_max}$ | 0.02±0.00 | 0.03±0.00 | 0.02±0.00 | 0.03±0.00 | 0.03±0.00 | **0.08±0.03** |
| $\mathcal{O}^{naive}_{plain}$ | 0.29±0.01 | 0.29±0.01 | 0.26±0.02 | 0.30±0.01 | 0.30±0.01 | **0.31±0.05** |
| size ratio | 3.070 | 3.110 | 3.060 | **2.89** | 2.95 | 15.640 |
| size std | 8.390 | 7.94 | 8.680 | **7.28** | 8.050 | 18.680 |

Table 9: Performance comparison on *Lead-Lag*. Results in each year is averaged over ten runs. Mean and standard deviation (after ±) are calculated over the 19 years. The best is marked in **bold red** and the second best is marked in underline blue. InfoMap results are omitted here as it usually predicts a single huge cluster and could not generate imbalance results.

| Metric/Method | Bi_sym | DD_sym | DISG_LR | Herm | Herm_rw | DIGRAC |
|---|---|---|---|---|---|---|
| $\mathcal{O}^{sort}_{vol\_sum}$ | 0.07±0.01 | 0.07±0.01 | 0.07±0.01 | 0.07±0.02 | 0.07±0.02 | **0.15±0.03** |
| $\mathcal{O}^{sort}_{vol\_min}$ | **0.51±0.10** | 0.48±0.09 | 0.47±0.10 | **0.51±0.11** | 0.50±0.10 | 0.47±0.09 |
| $\mathcal{O}^{sort}_{vol\_max}$ | 0.07±0.01 | 0.06±0.01 | 0.06±0.01 | 0.07±0.01 | 0.07±0.01 | **0.14±0.03** |
| $\mathcal{O}^{sort}_{plain}$ | **0.66±0.09** | 0.64±0.08 | 0.63±0.08 | **0.66±0.09** | 0.65±0.09 | 0.53±0.09 |
| $\mathcal{O}^{std}_{vol\_sum}$ | 0.04±0.01 | 0.04±0.01 | 0.04±0.01 | 0.04±0.01 | 0.04±0.01 | **0.12±0.03** |
| $\mathcal{O}^{std}_{vol\_min}$ | 0.27±0.04 | 0.27±0.04 | 0.25±0.04 | 0.27±0.03 | 0.27±0.03 | **0.38±0.07** |
| $\mathcal{O}^{std}_{vol\_max}$ | 0.04±0.00 | 0.03±0.00 | 0.03±0.00 | 0.03±0.00 | 0.03±0.00 | **0.11±0.02** |
| $\mathcal{O}^{std}_{plain}$ | 0.40±0.05 | 0.39±0.05 | 0.38±0.05 | 0.40±0.05 | 0.40±0.05 | **0.44±0.07** |
| $\mathcal{O}^{naive}_{vol\_sum}$ | 0.03±0.01 | 0.03±0.01 | 0.03±0.01 | 0.03±0.01 | 0.03±0.01 | **0.08±0.04** |
| $\mathcal{O}^{naive}_{vol\_min}$ | 0.20±0.05 | 0.19±0.05 | 0.18±0.05 | 0.19±0.04 | 0.19±0.04 | **0.26±0.10** |
| $\mathcal{O}^{naive}_{vol\_max}$ | 0.03±0.01 | 0.02±0.01 | 0.02±0.01 | 0.02±0.00 | 0.02±0.00 | **0.08±0.03** |
| $\mathcal{O}^{naive}_{plain}$ | 0.30±0.06 | 0.28±0.06 | 0.27±0.06 | 0.29±0.05 | 0.29±0.05 | **0.32±0.11** |
| size ratio | 3.67 | **3.34** | 3.900 | 4.110 | 3.880 | 8.070 |
| size std | 9.31 | **9.14** | 10.090 | 10.490 | 10.360 | 17.060 |

Table 10: Performance comparison on *Lead-Lag*, where we evaluate the performance distance to the best one in each year. Results in each year is averaged over ten runs. Mean and standard deviation (after ±) are calculated over the 19 years. The best is marked in **bold red** and the second best is marked in underline blue. InfoMap results are omitted here as it usually predicts a single huge cluster and could not generate imbalance results.

| Metric/Method | Bi_sym | DD_sym | DISG_LR | Herm | Herm_rw | DIGRAC |
|---|---|---|---|---|---|---|
| $\mathcal{O}^{\text{sort}}_{\text{vol\_sum}}$ | 0.07±0.02 | 0.08±0.02 | 0.08±0.02 | 0.07±0.02 | 0.07±0.02 | **0.00±0.00** |
| $\mathcal{O}^{\text{sort}}_{\text{vol\_min}}$ | **0.01±0.01** | 0.05±0.03 | 0.06±0.03 | 0.02±0.02 | 0.02±0.02 | 0.06±0.04 |
| $\mathcal{O}^{\text{sort}}_{\text{vol\_max}}$ | 0.07±0.02 | 0.07±0.02 | 0.07±0.02 | 0.07±0.02 | 0.07±0.02 | **0.00±0.00** |
| $\mathcal{O}^{\text{sort}}_{\text{plain}}$ | **0.01±0.02** | 0.03±0.03 | 0.05±0.03 | **0.01±0.02** | 0.02±0.02 | 0.14±0.03 |
| $\mathcal{O}^{\text{std}}_{\text{vol\_sum}}$ | 0.08±0.02 | 0.08±0.02 | 0.08±0.02 | 0.08±0.02 | 0.08±0.02 | **0.00±0.00** |
| $\mathcal{O}^{\text{std}}_{\text{vol\_min}}$ | 0.10±0.05 | 0.11±0.04 | 0.13±0.05 | 0.11±0.05 | 0.11±0.05 | **0.00±0.00** |
| $\mathcal{O}^{\text{std}}_{\text{vol\_max}}$ | 0.07±0.02 | 0.08±0.02 | 0.08±0.02 | 0.08±0.02 | 0.08±0.02 | **0.00±0.00** |
| $\mathcal{O}^{\text{std}}_{\text{plain}}$ | 0.04±0.03 | 0.05±0.04 | 0.06±0.04 | 0.04±0.04 | 0.04±0.03 | **0.00±0.00** |
| $\mathcal{O}^{\text{naive}}_{\text{vol\_sum}}$ | 0.05±0.03 | 0.06±0.03 | 0.06±0.03 | 0.05±0.03 | 0.05±0.03 | **0.00±0.00** |
| $\mathcal{O}^{\text{naive}}_{\text{vol\_min}}$ | 0.06±0.07 | 0.07±0.06 | 0.08±0.07 | 0.07±0.08 | 0.07±0.08 | **0.00±0.00** |
| $\mathcal{O}^{\text{naive}}_{\text{vol\_max}}$ | 0.05±0.03 | 0.05±0.03 | 0.05±0.03 | 0.05±0.03 | 0.05±0.03 | **0.00±0.00** |
| $\mathcal{O}^{\text{naive}}_{\text{plain}}$ | 0.03±0.06 | 0.05±0.05 | 0.06±0.06 | 0.04±0.06 | 0.04±0.06 | **0.01±0.02** |
| size ratio | 1.04 | **0.71** | 1.270 | 1.480 | 1.250 | 5.440 |
| size std | 0.58 | **0.41** | 1.360 | 1.770 | 1.630 | 8.340 |

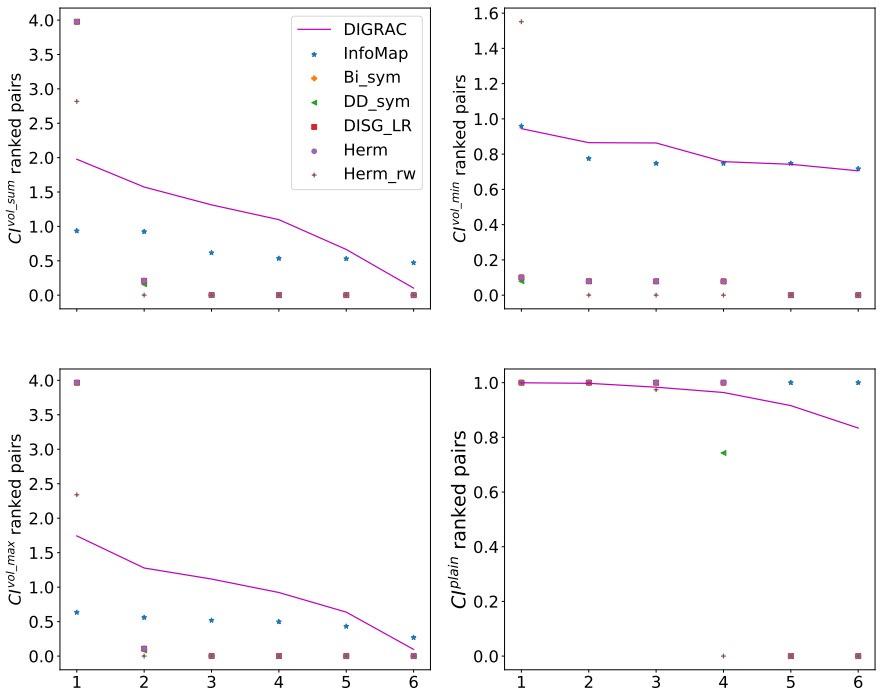

Figure 12: Ranked pairs of pairwise imbalance recovered by comparing methods for different choices of normalization on the *Telegram* data set. Lines are used to highlight DIGRAC's performance.

### D.2 RANKED PAIRWISE IMBALANCE SCORES

We also plot the ranked pairwise imbalance scores for all data sets except *Blog*, which has only one possible pairwise imbalance score. For *Lead-Lag*, we only plot the year 2015 as an example; the plots for the other years are similar. Figures 12, 13, 14 and 15 illustrate that DIGRAC is able to provide comparable or higher pairwise imbalance scores for the leading pairs, especially on CI$^{\text{vol\_min}}$ pairs. We also observe that except for CI$^{\text{plain}}$, DIGRAC has a less rapid drop in pairwise imbalance scores after the first leading pair compared to Herm and Herm_rw, which can have a few pairs with higher imbalance scores than DIGRAC.

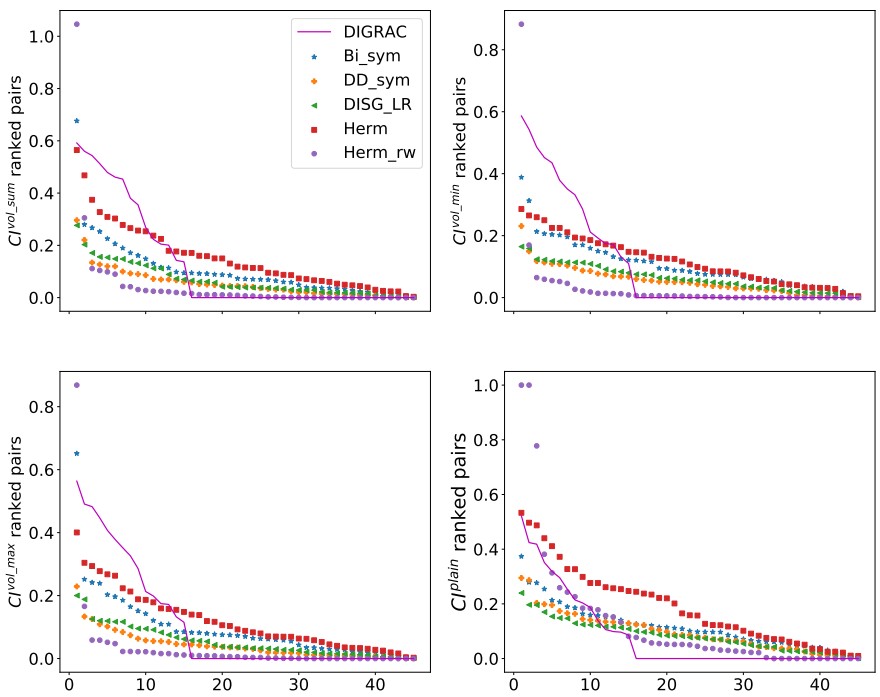

Figure 13: Ranked pairs of pairwise imbalance recovered by comparing methods for different choices of normalization on the *Migration* data set. Lines are used to highlight DIGRAC's performance. InfoMap results are omitted as it predicts one single huge cluster and could not produce imbalance results.

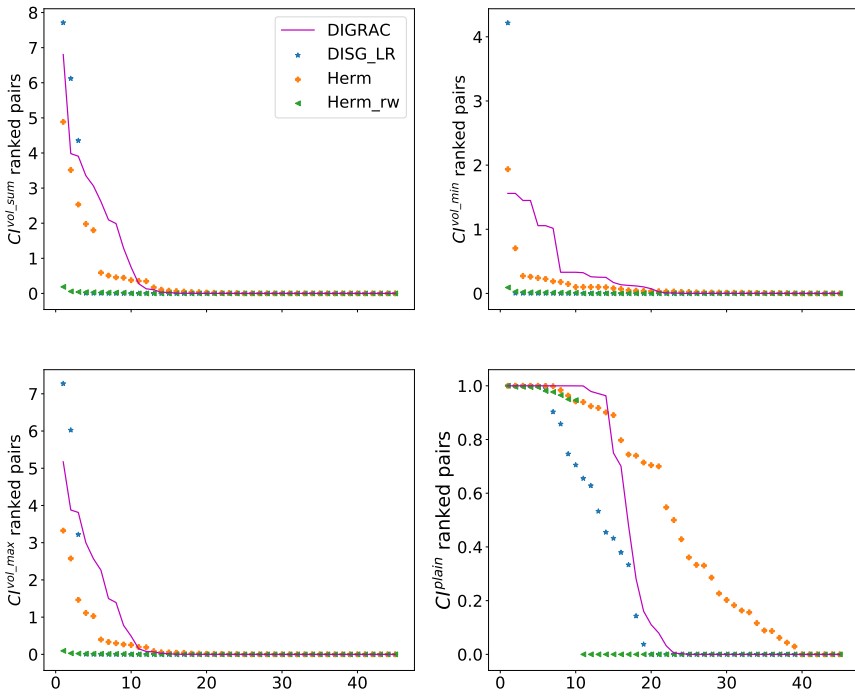

Figure 14: Ranked pairs of pairwise imbalance recovered by comparing methods for different choices of normalization on *WikiTalk* data set. Lines are used to highlight DIGRAC's performance. InfoMap results are omitted here because it triggers memory error due to the large number of predicted clusters.

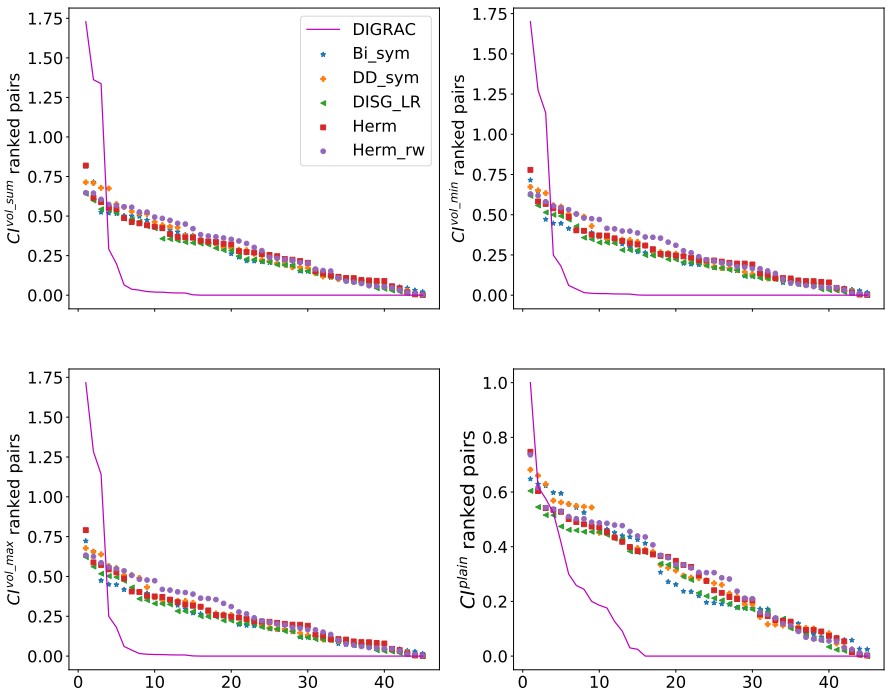

Figure 15: Ranked pairs of pairwise imbalance recovered by comparing methods for different choices of normalization on *Lead-Lag* data set. Lines are used to highlight DIGRAC's performance. InfoMap results are omitted here because it only predicts a single cluster.

### D.3 PREDICTED META-GRAPH FLOW MATRIX PLOTS

For each data set, we plot the predicted meta-graph flow matrix $\mathbf{F}'$ defined in Eq. (12).

From Figure 16, we conclude that DIGRAC is able to recover a directed flow imbalance between clusters in all of the selected data sets. Figure 16a shows a clear cut imbalance between two clusters, possibly corresponding to the Republican and Democratic parties. Figure 16b plots imbalance flows in the real data set *Telegram*, where cluster 3 is a core-transient cluster, cluster 0 is a core-sink cluster, cluster 2 is a periphery-upstream cluster, while cluster 1 is a periphery-downstream cluster (Elliott et al., 2020; Bovet & Grindrod, 2020). For *WikiTalk*, illustrated in Figure 16d, the lower-triangular part entries are typically source nodes for edges, while the upper-triangular part are target nodes. For *Lead-Lag*, taking the year 2015 as an example, DIGRAC is also able to recover high imbalance in the data.

We also note that DIGRAC would not necessarily predict the same number of clusters as assumed, so that we do not need to specify the exact number of clusters before training DIGRAC; specifying the maximum number of possible clusters suffices.

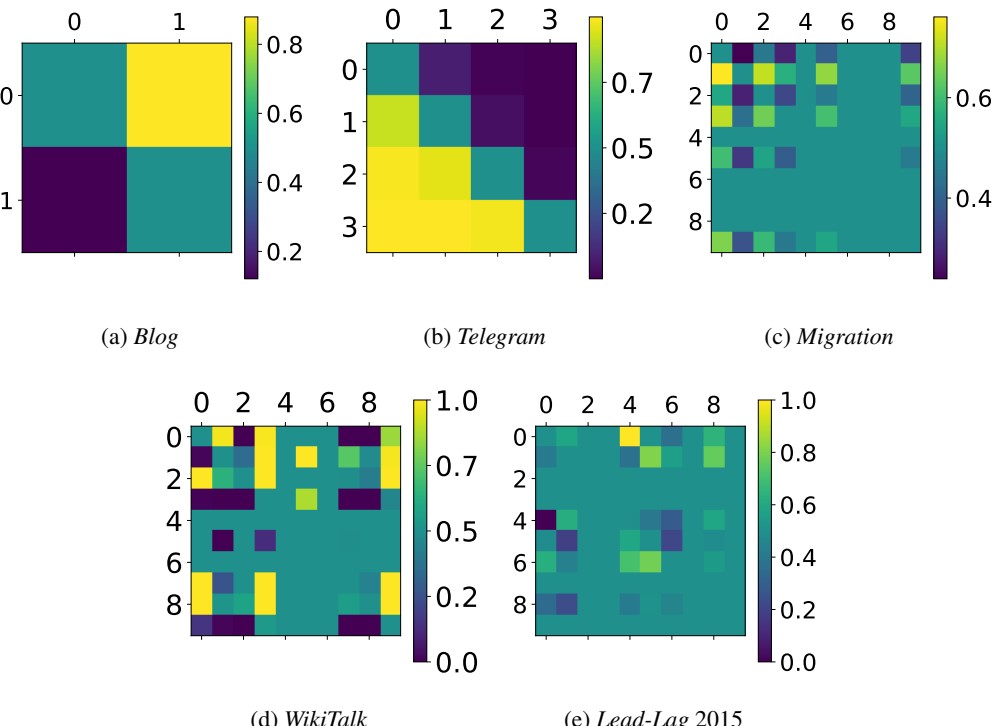

(a) *Blog*  (b) *Telegram*  (c) *Migration*

(d) *WikiTalk*  (e) *Lead-Lag* 2015

Figure 16: Predicted meta-graph flow matrix from DIGRAC of five real-world data sets.

## D.4 MIGRATION PLOTS

We compare DIGRAC to five spectral methods for recovering clusters for the US migration data set, and plot the recovered clusters on a map, in Figure 17.

The visualization in Figure (a-c) shows that clusters align particularly well with the political and administrative boundaries of the US states, as previously observed in Cucuringu et al. (2013). This outcome is not deemed too insightful, as it trivially reveals the fact there there is significant intra-state and inter-state migration, and does not uncover any of the information on latent migration patterns between far-away states, and more generally, between regions which are not necessarily geographically cohesive.

## D.5 COPING WITH OUTLIERS

As mentioned in Section B.3, the preprocessing step to use ratio of migration instead of absolute migration numbers is a way to cope with outliers (here, extremely large entries in the original digraph) in *Migration*. To validate the effectiveness of this approach to cope with outliers, Table 11 provides imbalance results for *Migration* when we do not transform the nonzero entries into ratios. Comparing with Table 6, we witness an overall decrease in the performance. In this case InfoMap no longer predicts a single huge cluster. However, its predicted number of clusters is about 44, which is too large. This also implies that InfoMap is very sensitive to the magnitude of digraph entries, while DIGRAC is not. Indeed, InfoMap gives 43 (too many) clusters for *Blog*, 19 (too many) for *Telegram*, 1 (too small) for *Migration*, and 17498 (far too many) for *WikiTalk*.

We compare DIGRAC to five spectral methods as well as InfoMap for recovering clusters for the US migration data set without the preprocessing step discussed earlier, and plot the recovered clusters on a map in Figure 19. Note that all methods, except DIGRAC, recover either clusters which are trivially small in size or contain one very large dominant cluster (as in (a), (b), (e) and to some extent, also (f)). The DISG_LR clustering and InfoMap clustering provide clear geographic boundaries, but

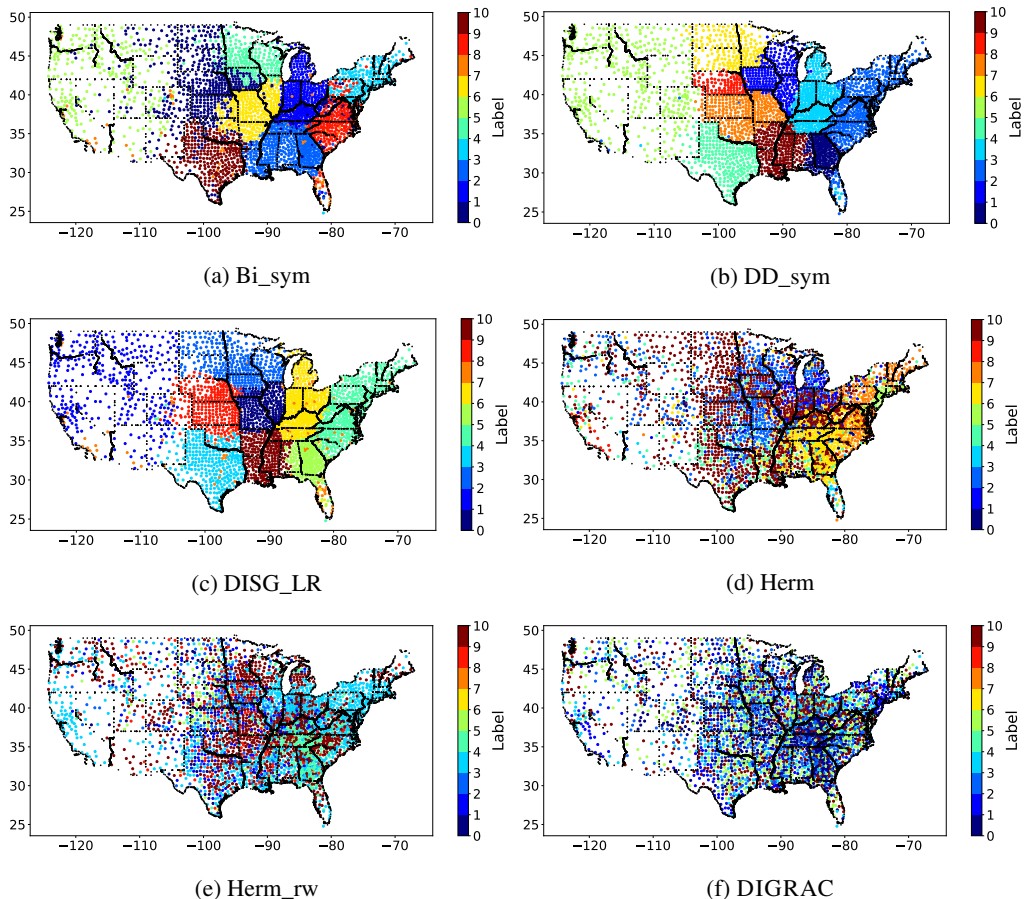

Figure 17: US migration predicted clusters, along with the geographic locations of the counties as well as state boundaries (in black). InfoMap results are omitted here because it only produces one huge cluster. The input data is normalized, following Eq. (13).

were not able to recover the imbalance among clusters. Other spectral methods generally have a dominant cluster containing most of the nodes, whereas DIGRAC has more balanced cluster sizes.

When employing methods that symmetrize the adjacency matrix (as in (a) and (b)), the migration flows between counties in different states will be lost in the process. Furthermore, the visualization in Figure (c) shows that clusters align particularly well with the political and administrative boundaries of the US states, as previously observed in Cucuringu et al. (2013). The same is for Figure (d). This outcome is not deemed very insightful, as it trivially reveals the fact that there is significant intra-state and inter-state migration, and does not uncover any of the information on latent migration patterns between far-away states, and more generally, between regions which are not necessarily geographically cohesive.

Figure 18 further plots the top three pairs of clusters based on four different imbalance scores given by DIGRAC. As shown in the figure, DIGRAC uncovers the migration trend from outside to inside, across states. This trend of the directed flow agrees with that discussed in Perry (2003), with many people migrating from New York and California to the inner states.

# E    DISCUSSION OF RELATED METHODS THAT ARE NOT COMPARED AGAINST IN THE MAIN TEXT

To further emphasize the importance of directionality, our synthetic data sets have no difference in density between clusters; their sole signal is in the directionality of the edges. If all edge directions

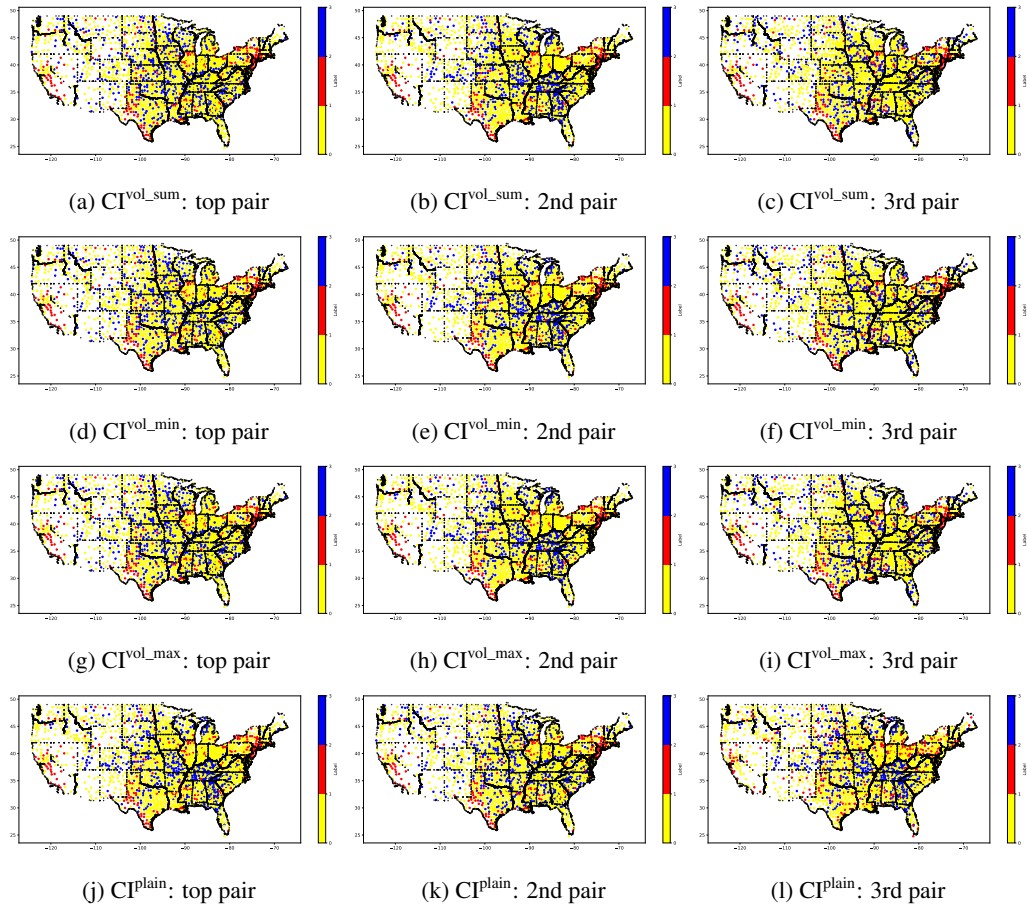

(a) $CI^{vol\_sum}$: top pair  (b) $CI^{vol\_sum}$: 2nd pair  (c) $CI^{vol\_sum}$: 3rd pair

(d) $CI^{vol\_min}$: top pair  (e) $CI^{vol\_min}$: 2nd pair  (f) $CI^{vol\_min}$: 3rd pair

(g) $CI^{vol\_max}$: top pair  (h) $CI^{vol\_max}$: 2nd pair  (i) $CI^{vol\_max}$: 3rd pair

(j) $CI^{plain}$: top pair  (k) $CI^{plain}$: 2nd pair  (l) $CI^{plain}$: 3rd pair

Figure 18: US migration predicted cluster pairs with top imbalance, along with the geographic locations of the counties as well as state boundaries (in black). Red (label 1) is the sending cluster while blue (label 2) is the receiving cluster. Yellow (label 0) denotes all the other locations being considered. Subcaptions show the imbalance score and the rank based on that score.

were to be removed, then no algorithm out there should be available to detect the clusters. To further support our claim why some methods mentioned in Section 2 in the main text are not appropriate for comparison, we have applied the default setting versions of the Louvain method (Dugué & Perez, 2015), the Leiden algorithm (Traag et al., 2019) and OSLOM (Lancichinetti et al., 2011), to our synthetic data sets, and find that they do not detect the structure in the data, with ARI and NMI values very close to zero, and very low imbalance values. In particular, Louvain and Leiden tend to give a larger number of clusters than the ground truth which is designed to have small cluster sizes. OSLOM outputs clusters with extreme sizes, either a huge cluster containing (almost always) all the nodes, or every node forming a cluster by itself.

On the real-world data sets, these methods often give numbers of clusters that do not match our expectations. (*Blog* has two underlying parties, *Telegram* has a four-cluster core-periphery structure). Louvain clusters nodes from *Blog* into 8-13 clusters (too many), *Telegram* into 4-5 clusters (acceptable), *Migration* into 5-7 clusters (acceptable), *WikiTalk* into 150-219 clusters (too many), and *Lead-Lag* into 10-55 clusters (acceptable or a bit too many). Leiden gives 12 (too many) clusters for *Blog*, 4-5 for *Telegram*, 5-6 for *Migration*, 170-248 (too many) for *WikiTalk*, and 10-55 clusters (acceptable or a bit too many) for *Lead-Lag*. OSLOM gives 6 clusters for *Blog* (too many), 16 for *Telegram* (too many), and 46 for *Migration* (too many). It could not generate results for *WikiTalk* after running for 12 hours, and hence we omit its discussion here. On *Lead-Lag*, OSLOM places every node in a single cluster for most of the years, and clusters the rest of the years into either a huge single cluster or two clusters.

Table 11: Performance comparison on *Migration* (without preprocessing). The best is marked in **bold red** and the second best is marked in underline blue.

| Metric/Method | InfoMap | Bi_sym | DD_sym | DISG_LR | Herm | Herm_rw | DIGRAC |
|---|---|---|---|---|---|---|---|
| $\mathcal{O}_{vol\_sum}^{sort}$ | 0.02±0.00 | 0.03±0.00 | 0.01±0.00 | 0.01±0.00 | **0.07±0.00** | 0.01±0.00 | 0.04±0.00 |
| $\mathcal{O}_{vol\_min}^{sort}$ | **0.24±0.00** | 0.20±0.01 | 0.12±0.02 | 0.14±0.00 | 0.21±0.01 | 0.05±0.02 | 0.18±0.02 |
| $\mathcal{O}_{vol\_max}^{sort}$ | 0.02±0.00 | 0.03±0.00 | 0.01±0.00 | 0.01±0.00 | **0.06±0.00** | 0.00±0.00 | 0.04±0.00 |
| $\mathcal{O}_{plain}^{sort}$ | 0.61±0.00 | 0.46±0.00 | 0.29±0.02 | 0.26±0.00 | **0.62±0.02** | 0.40±0.00 | 0.32±0.11 |
| $\mathcal{O}_{vol\_sum}^{std}$ | 0.00±0.00 | 0.01±0.00 | 0.00±0.00 | 0.00±0.00 | 0.02±0.00 | 0.00±0.00 | **0.03±0.01** |
| $\mathcal{O}_{vol\_min}^{std}$ | 0.03±0.00 | 0.09±0.00 | 0.04±0.01 | 0.05±0.00 | 0.08±0.01 | 0.02±0.01 | **0.11±0.03** |
| $\mathcal{O}_{vol\_max}^{std}$ | 0.00±0.00 | 0.01±0.00 | 0.00±0.00 | 0.00±0.00 | 0.01±0.00 | 0.00±0.00 | **0.02±0.01** |
| $\mathcal{O}_{plain}^{std}$ | 0.19±0.00 | 0.23±0.00 | 0.14±0.01 | 0.12±0.00 | **0.32±0.01** | 0.25±0.01 | 0.21±0.03 |
| $\mathcal{O}_{vol\_sum}^{naive}$ | 0.00±0.00 | 0.01±0.00 | 0.00±0.00 | 0.00±0.00 | 0.02±0.00 | 0.00±0.00 | **0.03±0.01** |
| $\mathcal{O}_{vol\_min}^{naive}$ | 0.02±0.00 | 0.08±0.00 | 0.04±0.01 | 0.05±0.00 | 0.08±0.01 | 0.02±0.01 | **0.11±0.04** |
| $\mathcal{O}_{vol\_max}^{naive}$ | 0.00±0.00 | 0.01±0.00 | 0.00±0.00 | 0.00±0.00 | 0.01±0.00 | 0.00±0.00 | **0.02±0.01** |
| $\mathcal{O}_{plain}^{naive}$ | 0.16±0.00 | 0.22±0.00 | 0.13±0.01 | 0.11±0.00 | **0.31±0.01** | 0.22±0.00 | 0.21±0.03 |
| size ratio | 8.500 | 3043.80 | 722.620 | 25.780 | **3059.20** | 415.880 | 203.230 |
| size std | **58.96** | 912.100 | 861.280 | 409.900 | 917.230 | 844.750 | 342.38 |

None of the methods outperform DIGRAC on our chosen performance measures from Table 1, except on the *Lead-Lag* data set. With regards to the 12 imbalance measures from SI Table 5, leaving out OSLOM as before, Louvain and Leiden perform poorly on all of the real data sets, except on *Lead-Lag*. Indeed, for *Lead-Lag*, the number of clusters we use for DIGRAC is ten according to the GICS sector memberships. However, if we use the sector memberships as labels, the imbalance values are poor, which implies that ten may not be a desirable choice of the number of clusters. Further, DIGRAC usually clusters the nodes into smaller number of clusters, while Louvain and Leiden usually cluster the nodes into a larger number of clusters (usually around 30, and sometimes above 50 clusters).

Table 12: Performance comparison on *Lead-Lag*, including Louvain and Leiden. Results in each year is averaged over ten runs. Mean and standard deviation (after ±) are calculated over the 19 years. The best is marked in **bold red** and the second best is marked in underline blue. InfoMap results are omitted here as it usually predicts a single huge cluster and could not generate imbalance results. Louvain and Leiden yield essentially identical results and often attain the highest objectives, while DIGRAC almost always places either first or second across all methods considered.

| Metric/Method | Louvain/Leiden | Bi_sym | DD_sym | DISG_LR | Herm | Herm_rw | DIGRAC |
|---|---|---|---|---|---|---|---|
| $\mathcal{O}_{vol\_sum}^{sort}$ | 0.08±0.02 | 0.07±0.01 | 0.07±0.01 | 0.07±0.01 | 0.07±0.02 | 0.07±0.02 | **0.15±0.03** |
| $\mathcal{O}_{vol\_min}^{sort}$ | 0.15±0.04 | **0.51±0.10** | 0.48±0.09 | 0.47±0.10 | **0.51±0.11** | 0.50±0.10 | 0.47±0.09 |
| $\mathcal{O}_{vol\_max}^{sort}$ | 0.08±0.02 | 0.07±0.01 | 0.06±0.01 | 0.06±0.01 | 0.07±0.01 | 0.07±0.01 | **0.14±0.03** |
| $\mathcal{O}_{plain}^{sort}$ | 0.15±0.04 | **0.66±0.09** | 0.64±0.08 | 0.63±0.08 | **0.66±0.09** | 0.65±0.09 | 0.53±0.09 |
| $\mathcal{O}_{vol\_sum}^{std}$ | **0.23±0.06** | 0.04±0.01 | 0.04±0.01 | 0.04±0.01 | 0.04±0.01 | 0.04±0.01 | 0.12±0.03 |
| $\mathcal{O}_{vol\_min}^{std}$ | **0.46±0.11** | 0.27±0.04 | 0.27±0.04 | 0.25±0.04 | 0.27±0.03 | 0.27±0.03 | 0.38±0.07 |
| $\mathcal{O}_{vol\_max}^{std}$ | **0.23±0.05** | 0.04±0.00 | 0.03±0.00 | 0.03±0.00 | 0.03±0.00 | 0.03±0.00 | 0.11±0.02 |
| $\mathcal{O}_{plain}^{std}$ | **0.46±0.11** | 0.40±0.05 | 0.39±0.05 | 0.38±0.05 | 0.40±0.05 | 0.40±0.05 | 0.44±0.07 |
| $\mathcal{O}_{vol\_sum}^{naive}$ | **0.23±0.06** | 0.03±0.01 | 0.03±0.01 | 0.03±0.01 | 0.03±0.01 | 0.03±0.01 | 0.08±0.04 |
| $\mathcal{O}_{vol\_min}^{naive}$ | **0.46±0.11** | 0.20±0.05 | 0.19±0.05 | 0.18±0.05 | 0.19±0.04 | 0.19±0.04 | 0.26±0.10 |
| $\mathcal{O}_{vol\_max}^{naive}$ | **0.23±0.05** | 0.03±0.01 | 0.02±0.01 | 0.02±0.01 | 0.02±0.00 | 0.02±0.00 | 0.08±0.03 |
| $\mathcal{O}_{plain}^{naive}$ | **0.46±0.11** | 0.30±0.06 | 0.28±0.06 | 0.27±0.06 | 0.29±0.05 | 0.29±0.05 | 0.32±0.11 |
| size ratio | 124.530 | 3.67 | **3.34** | 3.900 | 4.110 | 3.880 | 8.070 |
| size std | 47.960 | 9.31 | **9.14** | 10.090 | 10.490 | 10.360 | 17.060 |

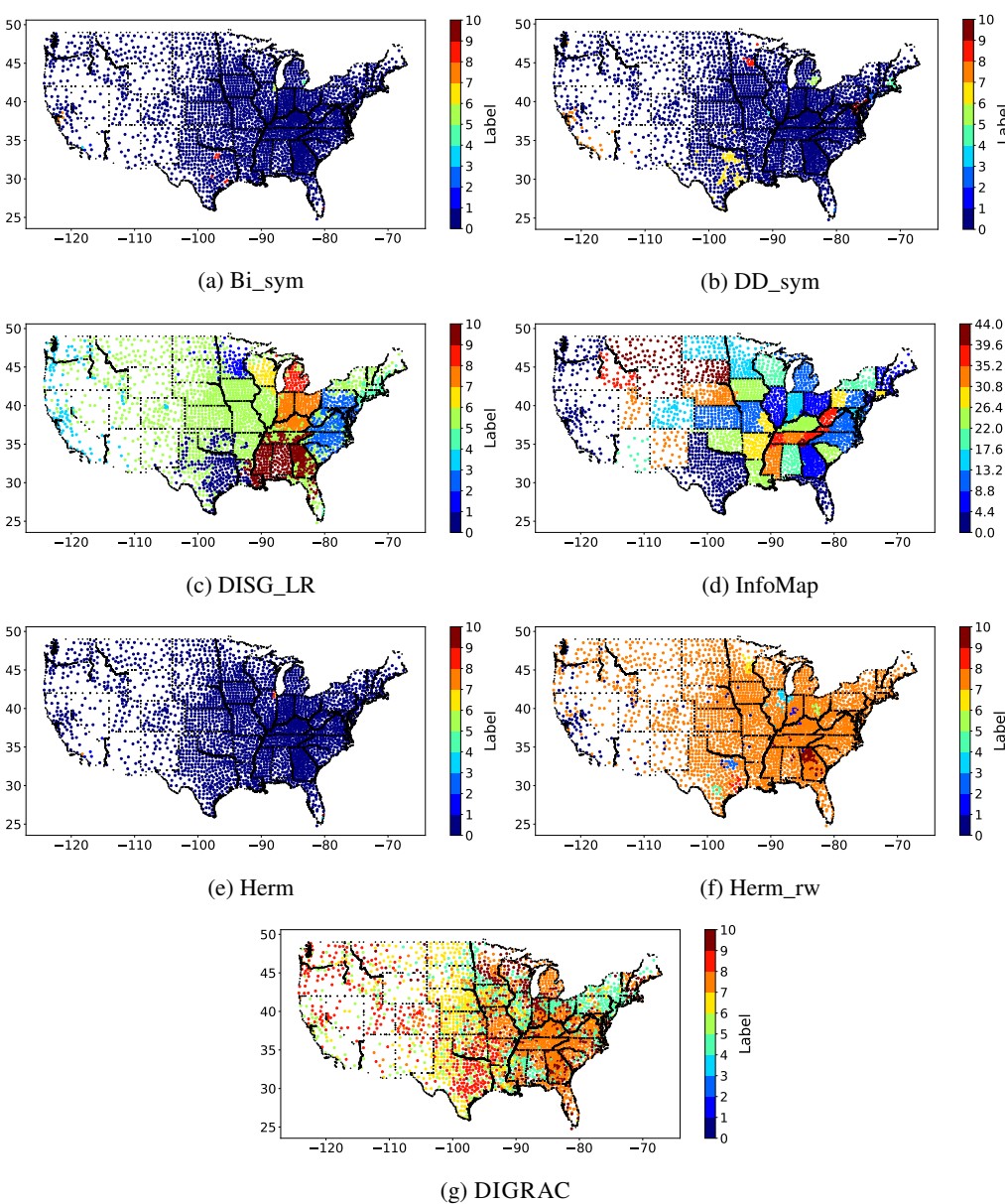

Figure 19: US migration predicted clusters, along with the geographic locations of the counties as well as state boundaries (in black). The input digraph has extremely large entries; unlike in Figure 17, we do not employ here the normalization given by Eq. (13). Altogether, this demonstrates the robustness of DIGRAC to outliers in the data, which is not a characteristic of other state-of-the-art methods such as Herm and Herm_rw.

