# OpenReview forum: " DIGRAC: Digraph Clustering Based on Flow Imbalance"
_ICLR.cc/2022/Conference — ICLR 2022 Submitted_

### Official Review · Reviewer_h5sc · 2021-11-02

**Correctness:** 3
**Technical Novelty And Significance:** 3
**Empirical Novelty And Significance:** 2
**Recommendation:** 5
**Confidence:** 4

**Main Review:**

Strengths:
- The paper provides an interesting pipeline for clustering in directed graphs. In particular, the idea of modeling the imbalance flow is very useful.
- Interesting benchmark graphs. I really liked the idea of using a modified DSBM model to express information flow.

Weaknesses:
- The theoretical contributions of the paper are somewhat vague at some points, and overall could be enhanced.
- It is not very clear what are the novel aspects of the DIMPA model.
- The baseline models could be enhanced.


**Detailed comments.**

Despite the interesting formulation of the problem, there are several aspects of the paper that are not quite clear and need further development. Next, I provide a detailed discussion of points that need to be addressed.

To begin with, despite having an interesting formulation of the imbalance flow, the loss function defined in Eq. (2) and (3), is not well-understood. I can see the semantics behind the imbalance term, but the theoretical properties are not clear at all. Is there any theoretical characterization for this loss? Are there cases where it could lead to degenerate solutions? It would also be important to clearly present the constraints of the optimization problem. Examining how the loss behaves in terms of convergence, is also important.

The other important point has to do with the novelty of the DIMPA algorithm. It’s not clear to me what are the novel parts of this model w.r.t. previous works. As far as I understand, the paper shares common ideas with message passing GNNs as well as with the KernelGCN model. Therefore, it’s difficult to evaluate its novelty w.r.t. state-of-the-art baseline models.

I really appreciated the effort that the authors put into the empirical analysis, especially on the parts related to the stochastic block model. Regarding the baseline models, the paper discusses why many “traditional” models are not part of the empirical analysis. Nevertheless, I would expect to add models that capture information flow, including for instance algorithms that rely on stochastic blockmodels (e.g., “Directed network community detection: A popularity and productivity link model”, SDM 2010). Besides, it would be interesting to have a model that is based on directed modularity (e.g., E. A. Leicht, M. E. J. Newman, “Community structure in directed networks”, PRL 2008).

Lastly, despite mentioning the complexity of the model, there are no scalability experiments with the running time of the algorithm and its comparison to baseline models.


**Summary Of The Paper:**

The paper introduces DIGRAC, an end-to-end model for clustering in directed graphs. The novelty of the model stems from the fact that it optimizes the directed flow imbalance to identify clusters that go beyond density-based criteria.  The proposed problem formulation is interesting: the loss function is defined in order to capture imbalance flow among clusters. Then, a GNN model is used to obtain the probabilistic assignment matrix that defines clusters. In terms of empirical analysis, the paper performs experiments on various artificial and real-world datasets, comparing the performance of the proposed model to various baselines.

**Summary Of The Review:**

The paper proposes an interesting approach to end-to-end flow-based clustering in directed graphs. Nevertheless, particular aspects of the paper, especially those related to the properties of the loss function and, partly, the experiments, need further development.

---

> ### Author Response · Authors · 2021-11-12
> **Theoretical Contributions and Discussion on DIMPA**
>
> Concern:
> The theoretical contributions of the paper are somewhat vague at some points, and overall could be enhanced. Despite having an interesting formulation of the imbalance flow, the loss function defined in Eq. (2) and (3), is not well-understood. I can see the semantics behind the imbalance term, but the theoretical properties are not clear at all. Is there any theoretical characterization for this loss? Are there cases where it could lead to degenerate solutions? It would also be important to clearly present the constraints of the optimization problem. Examining how the loss behaves in terms of convergence, is also important.
>
> Response:
> First, we would like to apologize; we should have merged the SI into the main submission document; the SI was instead as a separate .zip file. Hence the information in the SI was easy to overlook. Indeed, we have provided some theoretical analysis of the imbalance objective in SI A.
> The issue of degenerate cases is implicitly addressed in the paper. To avoid the degenerate case of zero in the denominater, we have added a small value to the denominator. When none of the imbalance values between pairwise clusters exceeds the threshold in the "std" variant, as detailed in SI Section A.3, we switch to the "naive" variant.
> We do not detail the degenerate solutions in the main part of the paper but our code, for which a link has also been provided, addresses these issues (specifically, starting from line 160 in https://anonymous.4open.science/r/1b728e97-cc2b-4e6a-98ea-37668813536c/src/metrics.py).
>
> As for constraints to the optimization problem, we require %entries of the adjacency matrix are nonnegative, the input to be probabilities,  and the number of clusters to be at least two. This constraint
> can be implied by the definitions in Section 3.1, as otherwise there is no imbalance "between" clusters but following your suggestion it is now explicitly included in Section 3.1.
> As for convergence, we compare the variants in SI A.4 empirically. During training we normally use early-stopping based on validation loss. Indeed, the loss may not converge or may not even decrease if we do not apply normalization, as implicitly indicated by the zero ARIs in SI A.4.
>
> Concern:
> It is not very clear what are the novel aspects of the DIMPA model. It’s not clear to me what are the novel parts of this model w.r.t. previous works. As far as I understand, the paper shares common ideas with message passing GNNs as well as with the KernelGCN model. Therefore, it’s difficult to evaluate its novelty w.r.t. state-of-the-art baseline models.
>
> Response:
> Indeed, as we have mentioned in our main text (the first paragraph on page 2, the sentence starting with "For the implementation of the framework"), DIMPA is not our main novelty, but is just one choice that could take directionality into account which is simple yet effective. We have made it clear that DIMPA is extended from KernelGCN for digraphs in the first paragraph of Section 3.2 in the main text. Other aggregation schemes are possible and welcome, as long as they take directionality into account; we would then apply our imbalance objective. To illustrate this point, we have validated the efficacy of our imbalance objective when using another aggregation method, DiGCN, in Figure 3(b) within Section 4.3 in the main text.

---

> ### Author Response · Authors · 2021-11-12
> **Baseline Models and Pointer to the SI .zip File**
>
> Concern:
> The baseline models could be enhanced. I really appreciated the effort that the authors put into the empirical analysis, especially on the parts related to the stochastic block model. Regarding the baseline models, the paper discusses why many “traditional” models are not part of the empirical analysis. Nevertheless, I would expect to add models that capture information flow, including for instance algorithms that rely on stochastic blockmodels (e.g., “Directed network community detection: A popularity and productivity link model”, SDM 2010).
>
>
> Response:
> Thank you for appreciating our effort. Again, apologies for not having merged the SI into the main submission document in our original submission but as a separate .zip file, which might have caused your having not seen those contents. Indeed, we have included several more baselines in SI E; we have not put them into the main text as we already have a large number of baseline models and those additional baselines perform poorly on our tasks as they do not view directionality as the main signal. For example, we have compared against Louvain and Leiden algorithms that are based on directed modularity, as well as against the OSLOM method. They are all discussed in our Related Work section (section 2), as well as the reasons why they are not included in the empirical analysis in the main text.
>
> The reasons why we have not included the probablistic model you mentioned that captures information flow are the following: 1) It is based on in-degree and out-degree, and the DISG\_LR method we have included in our experiments is also based on in-out degrees. 2) The InfoMap method that we have compared with in the main text is also based on information flow. 3) Due to page limit, we could not compare against all digraph methods, but we pick the most representative and relative methods in our experiments and we already have 11 state-of-the-art baselines in the main text and 3 in the SI. 4) This suggested paper does not provide code and does not cover initialization. While its original EM algorithm requires at least $O(n^2K)$ parameters $q_{ijk}$ for each iteration which is too expensive for large networks, the paper mentions that those complicated $q_{ijk}$ values are computed in an “on-demand” fashion, but no further details are provided.
> It is worth emphasizing that our synthetic data sets have no difference in density between clusters; their sole signal is in the directionality of the edges. If all edge directions were to be removed, then no algorithm out there should be available to detect the clusters.
>
>
> Concern:
> Lastly, despite mentioning the complexity of the model, there are no scalability experiments with the running time of the algorithm and its comparison to baseline models.
>
> Response:
> Indeed, we have briefly mentioned the total running time and some comparisons on the running time of different methods in SI B.2. Again, apologies for not having merged the SI into the main submission document in our original submission but as a separate .zip file, which might have caused your having not seen those contents. We would like to point out that none} of the baselines conduct experiments on very large-scale GNNs in their papers. In future work, we shall explore to follow  [Fey et al., 2021] in order to devise a more scalable version of DIGRAC.

---

### Official Review · Reviewer_mMXH · 2021-11-02

**Correctness:** 3
**Technical Novelty And Significance:** 2
**Empirical Novelty And Significance:** 2
**Recommendation:** 3
**Confidence:** 3

**Main Review:**

**Strengths**

Overall, the paper is well written.

**Weaknesses**

I am concerned with the practicality of the problem and consequently the method studied in this paper. In Section 4, the authors mention that "real data sets with ground-truth flow imbalances are not available to date", so they use normalized imbalance scores to evaluate clustering performance. This feels a bit like going against the natural order of how machine learning models are typically developed. Usually, we start with a real, practical problem at hand, and then we try and develop a new method to solve it, if existing methods do not yield satisfying results. However, I feel like this paper is going backwards: It starts with inventing a method that solves a possibly made-up problem. The experiments in Section 4 do not demonstrate the practical importance of clustering with respect to flow imbalances. Since this is an empirical paper, I believe that a solid empirical evidence that showcases the need for clustering with respect to flow imbalances is required.

More about empirical evaluations: The authors use 80% of all nodes for training. Isn't this percentage too much? Can't we use, e.g., 5% of all nodes for training?

**Summary Of The Paper:**

This paper studies the problem of clustering directed graphs where clustering structure is defined in terms of flow imbalances between clusters. The method proposed in this paper first computes node embeddings via directed mixed path aggregation, which takes directionality into account. In order to perform clustering, the node embeddings are transformed to an assignment probability matrix, which is then used to compute a flow imbalance score. The model is trained end-to-end to maximize the flow imbalance score (or its variants). Empirical experiments show that the proposed method has good performances.

**Summary Of The Review:**

I have concerns regarding both the problem setting and the empirical evaluation, so I do not view this paper as ready for publication at this time.

---

> ### Author Response · Authors · 2021-11-12
> **DIGRAC is not solving a made-up problem and has solid supporting evidence**
>
> Concern:
> I am concerned with the practicality of the problem and consequently the method studied in this paper. In Section 4, the authors mention that "real data sets with ground-truth flow imbalances are not available to date", so they use normalized imbalance scores to evaluate clustering performance. This feels a bit like going against the natural order of how machine learning models are typically developed. Usually, we start with a real, practical problem at hand, and then we try and develop a new method to solve it, if existing methods do not yield satisfying results. However, I feel like this paper is going backwards: It starts with inventing a method that solves a possibly made-up problem.
>
> Response:
>
> Thank you for the comments on our work.
> We believe the statement that we have \textit{invented a method that solves a possibly made-up problem}, is an unfair and inaccurate  evaluation of our work.
>
> Detecting flow patterns in directed networks is a problem which appears in so many applications that it warrants methods for detecting it.
> We have motivated the use of flow imbalance in the introduction section in the main text, including some real-world data sets in which flow imbalance is a major signal.
> For example, for the Telegram data in [Bovet and Grindrod, 2020] used in some of the experiments, the edge density within clusters derived in [Bovet and Grindrod, 2020] to model directed core-periphery structure, could even be smaller than the edge density across clusters, implying that uncovering  clusters based on edge density is misleading. The U.S. migration data is another example. If the goal is to uncover migration patterns among states, one needs to investigate migration flows across states instead of within state. Furthermore, flow imbalance can be used, for example, for lead-lag detection in time series data. Lead-lag relationships in multivariate time series are of interest in fields such as
>
> - biology:
> J. Runge, P. Nowack, M. Kretschmer, S. Flaxman, and D. Sejdinovic. Detecting causal associations in large nonlinear time series datasets. Science Advances, 5(11), 2019.
>
> - finance:
> D. Sornette and W. X. Zhou. Non-parametric determination of real-time lag structure between two time series: The ’optimal thermal causal path’ method. Quantitative Finance, 5(6):577–591, 2005.
>
> - earth sciences:
> A. Harzallah and R. Sadourny. Observed lead-lag relationships between Indian summer monsoon and some meteorological variables. Climate Dynamics, 13(9):635–648, 1997.
>
> The following lines of work, that have appeared at top venues in the last two years, address the very same problem we consider in our present work, of capturing imbalanced flows in directed graphs:
>
> [1] Hermitian matrices for clustering directed graphs: insights and applications Mihai Cucuringu, Huan Li, He Sun, Luca Zanetti (AISTATS 2020)
>
> [2] MagNet  - Xitong Zhang, Yixuan He, Nathan Brugnone, Michael Perlmutter, Matthew Hirn (NeurIPS 2021)
>
> [3] Higher-order spectral clustering of directed graphs.
> Steinar Laenen, He Sun (NeurIPS 2020).
>
> Furthermore, the usefulness of this methodology and flow imbalance-based concepts  were illustrated in this recent work for lead-lag detection in time series data:
>
> [4] Detection and clustering of lead-lag networks for multivariate time series with an application to financial markets,
> Stefanos Bennett, Mihai Cucuringu and Gesine Reinert, KDD Workshop on mining and learning from time series (2021)
> In light of such prior work, we humbly disagree with your statement evaluating our work and claiming that DIGRAC addresses a made-up problem, and find that it lacks any solid supporting evidence.

---

> > ### Comment · Reviewer_mMXH · 2021-11-30
> > **Thank you for your response**
> >
> > Dear author(s), thank you for your detailed and informative response. I am aware of some previous work you kindly mentioned, e.g., [1] and [3], which introduced some notable theoretical contributions. Given that theoretical analysis is not the main focus of this paper, and judged from an empirical perspective, I am not entirely convinced that the current version has sufficient empirical (or methodological) contributions. In particular, and I fully understand that you may have a different perspective and thus disagree, the lack of ground-truth flow imbalances in real datasets makes it a bit hard to appreciate the practical importance of the proposed method in a machine learning setting. For example, since DIGRAC tries to maximize the imbalance scores, the results in Table 1 is not surprising. While I do find this work interesting, I think that the paper in its current state might be a better fit for an applied data science venue.

---

> > > ### Author Response · Authors · 2021-11-30
> > > **Perhaps there has been a misunderstanding of the significance of the problem and DIGRAC's performance**
> > >
> > > Dear reviewer, the main focus of DIGRAC is the introduction of the use of an effective self-supervised imbalance objective which has been ignored by many methods especially modularity-based ones. It is not a paper driven by empirical analysis, and its has adequate theoretical contributions, which are mostly in SI due to space limitation.  Indeed, we have provided some theoretical analysis of the imbalance objective in SI A. In terms of empirical contributions, we have compared against over ten state-of-the-art methods on digraphs and show superior performance. As for methodological contribution, we have devise novel imbalance-based objectives with several variants, also with theoretical analysis. The lack of ground-truth flow imbalances in real-world datasets indeed implies our novelty. We would set DIGRAC as a benchmark and provide ground-truth imbalances for future researches. The problem itself is of great significance as we responded earlier, and that we try to draw people's attention to the novel aspect of clustering digraphs based on flow imbalance. The results in Table 1 are not all from imbalance objectives that DIGRAC has been trained on, and DIGRAC also performs great in synthetic data when ground-truth is available, in terms of both ARI and NMI values. We would appreciate it if you could kindly re-evaluate our work and try to understand the problem and results better.

---

> > > ### Author Response · Authors · 2021-12-01
> > > **Confusion by the Inconsistency of Your Reviews**
> > >
> > > Dear reviewer,
> > > You mentioned in our review that our paper “starts with inventing a method that solves a possibly made-up problem”, but then go to acknowledge familiarity with the above works [1] and [3]. We are honestly having a very hard time reconciling these two statements, given that our paper, before considering any real data sets, it first tackles the problem of recovering the clusters under a directed stochastic block model (DSBM). Such a DSBM model was considered under both [1] (general meta-graph setting) and [3] (under the very particular case of a path meta-graph). Our work is aiming to propose a GNN-based method to solve this problem, very much in the spirit of [2] (which also has no real theoretical guarantees on performance, the theorems listed in the supplementary material are standard results from the spectral literature or straightforward translations to the complex case), and whose performance is below that of our proposed method.
> > >
> > > [1] Hermitian matrices for clustering directed graphs: insights and applications Mihai Cucuringu, Huan Li, He Sun, Luca Zanetti (AISTATS 2020) [2] MagNet - Xitong Zhang, Yixuan He, Nathan Brugnone, Michael Perlmutter, Matthew Hirn (NeurIPS 2021) [3] Higher-order spectral clustering of directed graphs. Steinar Laenen, He Sun (NeurIPS 2020).

---

> > > > ### Comment · Reviewer_mMXH · 2021-12-01
> > > > **Thank you for raising your confusion. I believe the review is consistent throughout. Here are more details.**
> > > >
> > > > Dear author(s), thank you for the comment. Generally, I don't think it's the best practice to justify an acceptance by comparing with previously accepted papers. When I read this paper, I tried to evaluate its quality by its content. I think it is well written, I like the idea, but I am concerned with the practicality and applicability as I mentioned in the initial review. Since you specifically expressed your confusion with references to some recent work, let me try and explain a bit. (Again, it is not my intension to compare the contribution of this paper with accepted ones during a review process.) As a reader, when I read this paper and asked myself, are there enough contributions in other aspects which overweigh a possible lack of applicability? Based on the experiment with the real datasets, my answer was no. In the referenced papers you mentioned, either the idea is likely more novel at the time, or the algorithmic or spectral analysis is more interesting, or the experiments is more conniving as some real datasets have ground-truth labels. DIGRAC is not the first method that performs clustering on directed graphs, therefore I expected more than just an introduction to a new method. Repeating similar experiments on synthetic DSBM is fine, but as a reader I would like to see some real applicability. Is flow imbalance justifiably important? Does DIGRAC actually improve some real learning tasks, as opposed to synthetic or contrived examples? In my opinion, the empirical section did not convey a clear answer to these questions. For example, since the real datasets don't have ground-truth labels, it is hard to conclude that DIGRAC is really better than other methods. A more compelling metric (e.g. other than flow imbalance scores) may be required. I believe another reviewer also mentioned about the need for a more compelling evaluation in order for the readers to draw more meaningful conclusions, which I also agree with.

---

> > > > > ### Author Response · Authors · 2021-12-01
> > > > > **We believe there has been a misunderstanding of our contributions and hope that you could re-evaluate our work**
> > > > >
> > > > > Dear reviewer,
> > > > >
> > > > > First of all, linking to recently published papers is trying to convince you DIGRAC is not addressing a made-up problem and this imbalance aspect is praised. Further, this validates the practicality and applicability, as we also mentioned in our initial response. Comparing with recently published works is also a way of showing our compatibility with the ICLR community.
> > > > >
> > > > > The idea in itself is novel, as we mention in our introduction section of the main text.
> > > > >
> > > > > We have also mentioned why there is a lack of ground-truth for real-world data sets, which also motivated us to provide benchmarking evaluation metrics with theoretical analysis and empirical evidence.
> > > > >
> > > > > DIGRAC is not the first method on digraph clustering, but it is the first one based on the imbalance aspect using GNNs. There is no prior work introducing our imbalance objectives that are differentiable and could be trained using deep learning. Besides, we do not only introduce a new method, but we compare against and discuss previous related work in details.
> > > > >
> > > > > Also, DSBM experiments are not repeated, and the definition for the DSBM models in our setting is also novel, as no prior work has done the same thing, though our DSBM data could be viewed as an extension from theirs.
> > > > >
> > > > > As for real applicability, as we mentioned in our response to another reviewer, we have migration plots, fitted meta-graph plots etc., and we have provided a lot of evidence in the introduction part. And yes, DIGRAC improves no real tasks such as uncovering migration patterns from top imbalance pairs, which is also consistent with a survey for the patterns as mentioned in SI.
> > > > >
> > > > > We are better in terms of both imbalance values, and visual reconstruction results (this is hard to say which is the best, but visually we attain desirable performance). Our extensive experiments and plots should be able to convey a clear answer to all your concerns.
> > > > >
> > > > > Hope that you could understand our contributions as well as the problem better and re-evaluate our submission. Thank you!

---

> ### Author Response · Authors · 2021-11-12
> **DIGRAC is self-supervised in principle without label supervision and attains desirable performance**
>
> Concern: The experiments in Section 4 do not demonstrate the practical importance of clustering with respect to flow imbalances. Since this is an empirical paper, I believe that solid empirical evidence that showcases the need for clustering with respect to flow imbalances is required.
>
> Response:
> DIGRAC is not a purely empirical paper but rather a methods paper which has some theoretical contributions, see for example SI A, and which also tackles an important problem  in real-world data sets.
>
> Our empirical results show that clustering methods which are based on density, such as those maximizing directed modularity, cannot uncover clusters which are driven by flow structure in the meta-graph, see SI E (the SI is submitted via a .zip file in our original submission), unless they have a clear edge density signal as well.
>
> Moreover, DIGRAC outputs can be subsequently incorporated in a time series prediction pipeline based on lead-lag relations, such as the above reference [4].  Hence DIGRAC is not only a stand-alone method but can also be used in combination with other methods.
>
> All the above aspects support the claim that DIGRAC is not driven by a made-up problem, and that detecting this aspect of flow imbalance is important, and that there is an abundance of empirical evidence which shows the need for clustering with respect to flow imbalance.
>
> Concern:
> More about empirical evaluations: The authors use 80% of all nodes for training. Isn't this percentage too much? Can't we use, e.g., 5% of all nodes for training?
>
> Response:
> Thank you for your question. There may be some misunderstanding about the training process. We use 0% of the labels in training.  As DIGRAC is a self-supervised method, in principle, we could use all nodes for training. However, for a fair comparison with other GNN methods we use only 80% of the nodes for training. For supervised methods our split of 80% - 10% - 10% is a standard split. For the non-GNN methods, all nodes are used for training.
> This has been mentioned in the paragraph right before Section 4.1 in the main text, the abstract, and the first paragraph of SI D.1.
> We have modified the main text and the text in SI B.3 to further emphasize this choice.
>
> It is true that 80% is rather high for label supervision, but this choice was made to give other GNN methods an almost ideal situation. The other GNNs usually attain worse performance compared to DIGRAC even given such a high percentage of label supervision. This in turn further supports DIGRAC's superiority.
>
> As part of the ablation study, we also conduct experiments on DIGRAC using some label information, in Figure 3(d) of Section 4.3 in the main text.
> Here, varying the ratio of seed nodes for which labels are known during training (we take 0, 10, 50 and 100 % of the training nodes to have known labels, respectively), shows that DIGRAC
> can be turned into a supervised method by also taking known labels into account, as do the GNNs it is compared to. In this situation, DIGRAC performs even better.

---

### Official Review · Reviewer_SM6X · 2021-11-03

**Correctness:** 4
**Technical Novelty And Significance:** 3
**Empirical Novelty And Significance:** 3
**Recommendation:** 5
**Confidence:** 3

**Details Of Ethics Concerns:**

I don't think the paper introduces new ethical concerns.

**Main Review:**

**Summary** This paper presents an empirical study of clustering nodes based on a given graph structure. The clustering model is a deep approach that produces embeddings of the nodes that can be used for other tasks like node classification, etc.

**Merits**. This paper demonstrates that it can achieve the desired goal of node representations with the DIGRAC architecture and the "DIRECTED MIXED PATH AGGREGATION (DIMPA)" layer in particular. To better understand this deep model, the authors provide synthetic empirical results, which demonstrate the performance of the method under different kinds of data.

**Concerns**. My primary concern for this paper is that I find it difficult to understand the empirical performance gains provided by the method. In many of the ARI graphs on synthetic data, it seems many of the methods are quite close together in performance. In the real data, it seems that methods overlap in their error bars. I am also wondering whether more can be said about when DIGRAC's model for clustering is appropriate for a dataset and when it is not. And how adaptive the model can be in terms of performance despite those considerations. Apologies if I have misunderstood the empirical results, if so, I would like to better understand them.

**Summary Of The Paper:**

This paper introduces DIGRAC, a novel GNN approach for clustering based on the edges that are not present in a digraph. The architecture achieves this by finding node embeddings for digraphs that directly maximizes flow imbalance between clusters. The paper presents empirical results on both real and synthetic datasets, showing the effectiveness of the proposed approach.

**Summary Of The Review:**

An interesting paper with a self-supervised novel architecture for node clustering using an objective based on missing edges in a digraph. There are some concerns regarding the empirical gains provided by the method.

---

> ### Author Response · Authors · 2021-11-10
> **Explanation on Empirical Performance and Pointer to SI .zip File**
>
> Thank you for finding our work novel and interesting. Here we provide an explanation to address your concerns.
>
> First, we would like to apologize; we should have merged the SI into the main submission document instead of submitting it as a separate .zip file. Hence the information in the SI was easy to overlook.
>
> It is difficult to see the distinction of the performance because we start from the zero-noise regime, in which several methods perform almost equally well, to the very high noise regime, in which all methods perform poorly.  However, DIGRAC is among the best-performing methods, across all synthetic data sets considered. Moreover, the numerical results for the real-world data sets are showcased in Table 1, illustrating that DIGRAC performs best in all eight scenarios (across four data sets and two objective function evaluations.
> This empirical performance is rather remarkable, because DIGRAC is trained without any label supervision, while the other GNNs from the directed graph literature require labels for all training data (as it has been mentioned in the paragraph right before Section 4.1 in the main text, the abstract, and the first paragraph of SI D.1. We will also modify the main text to further emphasize this). Obtaining comparable or even better performance than fully supervised GNN methods, in a self-supervised manner, with a simple yet effective DIMPA aggregation scheme, serves as validation of DIGRAC's efficacy and provides ground for the novelty and soundness of our proposed methodology. In the paper, we have also included detailed discussions of some specific scenarios where some other GNNs perform the best, as detailed in SI C.2 (SI is submitted via a .zip file in our original submission), due to the use of the proximity neighborhood heuristic in DiGCN_ib, which aligns well with how the synthetic data is constructed.
>
> In detail, as stated in SI C.2, the multipartite, the cycle and the star settings correspond to the intuition behind DiGCN_ib which assumes that nodes are similar if their set of $k^{th}$-order neighborhoods are similar; here the second-order neighborhoods are similar by design. For networks with underlying meta-graph structure ``star", ``cycle" or ``multipartite", clusters could be determined by grouping nodes that share similar in-neighbors and out-neighbors, which aligns well with the second-order proximity used in DGCN and DiGCN_ib. Therefore, these methods are naturally well-suited for dealing with such synthetic data. We also note that although DIGRAC does not explicitly use second-order proximity, it can achieve comparable performance with DGCN and DiGCN_ib. This indicates DIGRAC's flexibility to adapt to directed networks with different underlying topologies, without explicitly utilizing higher-order proximity. In contrast to DIGRAC, DiGCN_ib is fully supervised, and takes much more space and time to implement, than DIGRAC.  This is partially due to the use of the so-called {\it inception blocks} in DiGCN_ib, where multi-scale directed structure features are encoded and fused with a fusion function. As stated in the original DiGCN paper by Tong et al. (NeurIPS2020), the worst space complexity is $\mathcal{O}(k'n^2),$ where $k'$ is the order of proximity to consider (we use $k'=2$ throughout). The eigenvalue decomposition in the preprocessing step is $\mathcal{O}(n^3).$ We also remark that the approximate Laplacian based on personalized PageRank, when no inception blocks are used, performs no better than the simpler implementation without the approximation.
>
> Overall DIGRAC is a fast method for the important problem of general directed clustering when directionality is the main signal. Despite its generality it performs at least as well as custom-tailored methods when the proximity neighborhood heuristic holds, while outperforming all tested methods on the complete meta-graph, where the proximity neighborhood  heuristic does not hold. We have performed a very comprehensive set of numerical simulations, on both synthetic and real data, and provided a fair comparison (in terms of various objective functions one may aim to maximize) against numerous SOTA methods, including methods (along with use-cases of their applicability) that have been well received by the recent ML community, and accepted to top venues such AISTATS, ICML, NeurIPS, KDD in the last two years. Therefore, we believe that DIGRAC is of considerable interest to the ML community.

---

### Author Response · Authors · 2021-11-12
**Updates on the submission pdf**

- We have merged SI (which was originally submitted in a separate .zip file) into the main submission file as appendix.

- We have explicitly emphasized that all methods (except the GNNs) that we compare DIGRAC against, are self-supervised methods; all baseline GNNs require label supervision. Furthermore, we clarified that DIGRAC is trained in a self-supervised manner without label supervision on the subgraph induced by the training nodes.

- We have added more details on the training setup in SI B.3.

- We have clearly described the constraint of having at least two clusters for the objective to hold in Section 3.1.

---

### Decision · Program_Chairs · 2022-01-20

**Decision:**

Reject

**Comment:**

The authors propose a new algorithm for clustering direct networks. The key idea behind the paper is to introduce a new flow imbalance measures and a new self-supervised GNN model to solve the task.

Overall, the paper is interesting and it introduces some new ideas although it needs additional work before being published.

In particular,
- the experiments could be improved by emphasizing more the evaluation on vol_sum/vol_max/etc metrics and by adding additional results on them
- the clarity of the experimental results should also be improved(for example, metrics / claims around Figure 4 still a bit hard to parse)
- finally, the paper would benefit by some theoretical results on the guarantees of the algorithm(most previous work in the area present interesting theoretical guarantees)